# Spectral Analysis of Molecular Features: When Richer Features Do Not Guarantee Better Generalization

## Abstract

The spectral properties of feature embeddings offer critical insights into model generalization and representation quality. While deep learning models are widely used for molecular property prediction, kernel methods remain competitive in low-data regimes, yet their spectral behavior is largely unexplored. We present the first comprehensive spectral analysis of kernel ridge regression across diverse representations—including molecular fingerprints (ECFP), pretrained transformers, graph neural networks, and 3D descriptors—evaluated on QM9 and 3 MoleculeNet benchmarks. Surprisingly, richer spectral features do not consistently yield better generalization performance, contradicting common representation heuristics used in self-supervised learning (SSL). Across 4 spectral metrics, only ECFP-based kernels show a strictly positive correlation with performance. Transformer and global 3D representations exhibit mixed behavior, whereas local 3D representations show consistently negative correlations. Truncation analysis further emphasizes this disparity: for local 3D representations on thermodynamic targets, fewer than 2% of eigenvalues (and occasionally as few as 0.02%) are needed to recover 95% of performance, whereas ECFP and transformer kernels require significantly more. By demonstrating a strong dependence on both task and representation, our results challenge the heuristic that richer spectra inherently improve generalization, providing new guidance for evaluating representations in SSL and in label-limited scientific tasks.

## 1 Introduction

Accurate molecular property prediction lies at the heart of modern materials-discovery pipelines, where the rapid estimation of chemical properties dramatically accelerates screening and design Bohacek et al. (1996); Reymond (2015); Goh et al. (2017); Kailkhura et al. (2019); Shen & Nicolaou (2019); Schapin et al. (2023); Kuang et al. (2024). Within this domain, two dominant modeling paradigms have emerged: neural network-based and kernel-based models. Neural networks (NNs) have advanced rapidly, driven by massive datasets and specialized architectures such as graph neural networks (GNNs) and pre-trained transformers Jiang et al. (2021); Le et al. (2022); Ju et al. (2023), giving latent feature representations for molecules Praski et al. (2025).

In contrast, traditional kernel methods excel in low-data regimes and rely on tailor-made kernel features implicitly defined by the kernel. Their non-parametric nature allows them to capture complex similarity structures without requiring massive training datasets or extensive hyperparameter tuning, making them especially valuable for sample-efficient tasks like active learning and Bayesian optimization Griffiths et al. (2023); Ralaivola et al. (2005); Bartók et al. (2013); Khan et al. (2023). Furthermore, kernel methods underpin some of the most successful machine-learned interatomic potentials Kamath et al. (2018); Vargas-Hernández & Gardner (2021); Thant et al. (2025), enabling accurate predictions of atomic forces and energies across diverse chemical systems. However, the design of these molecular kernels is challenging and highly varied: molecules can be represented using Cartesian or internal coordinates, cheminformatics descriptors like extended connectivity fingerprints (ECFPs), or complex graphs of atoms and bonds Griffiths et al. (2023).

Moreover, traditional evaluation of these molecular representations/embeddings (tailor-made kernel features or learnt latent features), involves solely on their test-set performance in downstream tasks. While pragmatic, this approach obscures deeper, fundamental questions regarding representation quality:

> *How well does a kernel capture the intrinsic structure of the target function,*
> *and what does this reveal about its generalization capacity?*

To peek inside this black box, machine learning theory points toward the *kernel spectrum*. Spurred by the theory of the neural tangent kernel (NTK) in over-parameterized neural networks Jacot et al. (2018), theoretical interest in the performance guarantees of kernel methods has surged, particularly concerning bounds dictated by spectral properties Arora et al. (2019); Mallinar et al. (2022); Li et al. (2023); Barzilai & Shamir (2024); Cheng et al. (2024b). Concurrently, the self-supervised learning (SSL) community has adopted spectral metrics to evaluate representation quality using unlabeled data Agrawal et al. (2022); Garrido et al. (2023). This has popularized a prevailing heuristic: one should choose the model with the richest feature spectrum, under the belief that "richer features yield better generalization."

**Contribution**  In this work, we investigate whether these theoretical insights regarding spectral richness hold true in the context of molecular chemistry. Our key contributions are summarized as follows:

- **Comprehensive spectral analysis of kernel and SSL features** We present the first systematic spectral analysis of molecular property prediction, evaluated across the QM9 dataset and three MoleculeNet benchmarks (ESOL, FreeSolv, and Lipophilicity), including $R^2$ score, four spectral metrics, and Pearson correlation between them. We also include an ablation study on the feature spectrum with artificially injected noise in the molecular features.
- **Kernel Probing.**  We also invent and apply Kernel Probing (KP)—kernel ridge regression on SSL features with linear probing (LP) as a special case—achieving improved performance over the commonly used LP baseline. This could be of independent for practitioners on SSL model evaluation and theorists on this hybrid approach of kernelized SSL features.
- **Truncated Threshold of Features.**  We extend the concept of *truncated kernels* Amini et al. (2022) to ECFP-based representations, quantifying the fraction of eigenvalues required to recover 95% and 99% of the original performance. Our results demonstrate that the top eigenvalues capture the vast majority of predictive power, further questioning the general belief that a long-tailed, richer spectrum is strictly necessary for generalization.

Table 1: Summary of molecular representations analyzed and the correlation between their spectral richness and predictive performance.

| Features | Computation | Type | Count | Correlation |
|---|---|---|---|---|
| **Kernel Features** | Defined by hand-crafted kernels | ECFP | 13 | Consistently Positive |
| | | 3D-Global | 3 x 3 | Mixed |
| | | 3D-Local | 3 x 2 | Consistently Negative |
| **Pre-Trained Features** | Learnt by pre-trained backbone | Transformer-based | 4 x 3 | Mixed |
| | | GNN-based | 2 x 3 | Mixed |

**Organization**  The paper is structured as follows. In Section 2, we review the relevant background and key concepts. Section 3 presents our experimental methodology and results. In Section 4, we discuss the novelty, limitations, and potential future directions of our work. Due to space constraints, additional experimental results, analyses, and discussions are provided in the Appendix.

## 2  Background

In molecular property prediction, the inputs are molecules $\mathcal{M}$, discrete objects without an inherent Euclidean representation. This necessitates the use of domain-specific representations, each inducing a corresponding kernel that encodes molecular similarity.

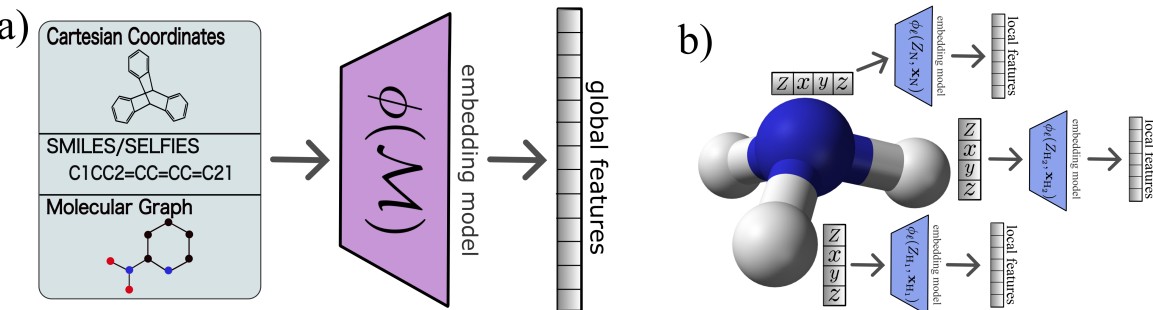

Figure 1: Molecular representation generation workflow compatible with kernel functions; (a) global ($\phi(\mathcal{M})$), where $\mathcal{M}$ represents a molecule, and (b) local ($\phi_\ell(Z_i, \mathbf{x}_i)$) molecular representations, $Z$ and $\mathbf{x}_i$ are the atomic number and coordinates, respectively, of the $i$th atom.

**Molecular Representations & Features**   In the machine learning literature, the terms *representation*, *feature*, and *embedding* are often used interchangeably. To avoid ambiguity, we define *molecular representations* as the raw formats used to encode molecules, such as 3D Cartesian coordinates, strings (SMILES or SELFIES), or molecular graphs. As illustrated in Fig. 1, representations are typically grouped into two categories: (i) *global descriptors*, which encode the entire molecule, and (ii) *local descriptors*, which capture individual atomic environments. In this work, **3D** kernels refer to those based on Cartesian coordinates (see Section B for details). And we define *(pre-trained) molecular features* as the vector embeddings extracted by passing a molecular representation through a pre-trained encoder or backbone $h$. For a given molecule $\mathcal{M}_i$, its pre-trained feature is denoted as $\mathbf{h}_i \stackrel{\text{def.}}{=} h(\mathcal{M}_i) \in \mathbb{R}^P$.

**Kernel Methods & Kernel Features**   On the other hand, a molecular kernel $k$ acts as a similarity measure mapping any two molecules $\mathcal{M}_i$ and $\mathcal{M}_j$ to a real number, forming a symmetric positive semi-definite kernel matrix $\boldsymbol{K} \in \mathbb{R}^{N \times N}$ over a dataset. By Mercer's theorem, any such kernel implicitly maps the input data into a reproducing kernel Hilbert space (RKHS) $\mathcal{H}$, allowing us to express the kernel as an inner product: $k(\mathcal{M}_i, \mathcal{M}_j) = \langle \phi(\mathcal{M}_i), \phi(\mathcal{M}_j) \rangle_{\mathcal{H}}$. Analogous to *pre-trained features*, we define *kernel features* as the vector $\boldsymbol{\phi}_i \stackrel{\text{def.}}{=} [\phi_p(\mathcal{M}_i)]_{p=1}^P \in \mathbb{R}^P$, where $\phi_p$ are the eigenfunctions of the kernel and $P$ can be infinite.

**Truncated Kernel Ridge Regression**   Standard kernel ridge regression (KRR) inherently biases toward eigenfunctions associated with larger eigenvalues Basri et al. (2020). To isolate the most informative components, truncated kernel ridge regression (TKRR) Amini et al. (2022) restricts the model to the top $r$ eigen-components. Given a kernel matrix with eigendecomposition $\mathbf{K} = \sum_{k=1}^N \mu_k \mathbf{u}_k \mathbf{u}_k^\top$, the rank-$r$ truncated kernel matrix is $\mathbf{K}^{(r)} = \sum_{k=1}^r \mu_k \mathbf{u}_k \mathbf{u}_k^\top$. To evaluate the TKRR predictor on a new test point $\mathcal{M}$, we use the approximated truncated kernel:

$$\tilde{k}^{(r)}(\mathcal{M}_i, \mathcal{M}) = [\mathbf{U}_{\leq r} \mathbf{U}_{\leq r}^\top \mathbf{k}]_i, \tag{1}$$

where $\mathbf{U}_{\leq r} = (\mathbf{u}_k)_{k=1}^r \in \mathbb{R}^{N \times r}$ and $\mathbf{k} = (k(\mathcal{M}_j, \mathcal{M}))_{j=1}^N \in \mathbb{R}^N$. This formulation effectively limits the predictor to a finite subset of *truncated kernel features* $\tilde{\boldsymbol{\phi}}^{(r)} \in \mathbb{R}^r$. Further theoretical properties of $\tilde{k}^{(r)}$ are detailed in Section E.

**Feature Spectrum**   To analyze the structural properties of these feature spaces, we examine their covariance. For pre-trained features stacked in matrix $\mathbf{H}$, the (empirical) feature covariance is $\hat{\boldsymbol{\Sigma}} = \frac{1}{N} \mathbf{H} \mathbf{H}^\top \in \mathbb{R}^{P \times P}$ which can be computed readily once the molecular features are extracted from the pre-trained backbone encoder. For (truncated) kernel features, the equivalent is the empirical covariance operator $\hat{\boldsymbol{\Sigma}} = \frac{1}{N} \Phi \Phi^\top \in \mathbb{R}^{P \times P}$, which has the same non-zero spectrum as the empirical kernel matrix $\frac{1}{N} \boldsymbol{K} \in \mathbb{R}^{N \times N}$. The eigenvalues of $\boldsymbol{\Sigma}$ constitute the *feature spectrum*. Intuitively, a richer feature spectrum means the feature vectors span more diverse directions in the ambient space, capturing finer details. In a kernel perspective, a faster spectral decay restricts the RKHS size, reducing kernel capacity. To quantify this richness, we arrange the empirical spectrum $\{\mu_1, \mu_2, \ldots, \mu_p\}$ in decreasing order. Assuming a power law $\mu_j \propto j^{-\alpha}$ Agrawal et al. (2022); Mallinar et al.

(2022), a smaller decay rate $\alpha$ indicates richer features. Alternative non-parametric measures of richness include spectral Shannon entropy (SSE) Huh et al. (2023); Garrido et al. (2023), intrinsic dimension (ID), and stable rank (SR) Ipsen & Saibaba (2024). These metrics are explicitly defined in Section D.

**Self-Supervised Learning & generalization**  In self-supervised learning (SSL), downstream property prediction is conventionally executed via linear probing—training a linear regressor with learned weights on the frozen pre-trained features $\mathbf{h}_i$. Crucially, $\ell_2$-regularized linear probing is mathematically equivalent to KRR using the linear kernel defined by the pre-trained features. Motivated by this equivalence, evaluating model quality via the feature spectrum has become standard practice in SSL Agrawal et al. (2022); Garrido et al. (2023). These SSL analyses, much like parallel studies on empirical kernel spectra Mallinar et al. (2022); Cheng et al. (2024a), rely heavily on the heuristic that "richer features yield better generalization." By unifying pre-trained and kernel features under this spectral framework, we rigorously test the validity of this heuristic within the domain of molecular chemistry.

## 3 Result

This section consists of (1) a comprehensive evaluation of the molecular property prediction as regression task over various molecular features, (2) a computation on 4 spectral metrics on each type of features together with a performance-feature correlation analysis, (3) an ablation test on the robustness of spectral metrics, and (4) a study of truncated threshold for truncated Kernel Ridge Regression, where a small portion of remaining molecular features ensure 95% of original performance.

### 3.1 Regression Performance

**Datasets**  In this paper, we consider the molecular property prediction as regression task on two types of molecular datasets: (a) **QM9** dataset Ramakrishnan et al. (2014) with 7 properties/labels; (b) ESOL, FreeSolv, and Lipophilicity **MoleculeNet benchmark** with 1 property each.

**Molecular Features as Models**  Our experiment involves kernel features implicitly defined by 13 ECFP-based kernels, 3 3D-global kernels and 3 3D-local kernels, and pre-trained features explicitly extracted by 4 transformer-based encoders and 1 GNN-encoder:

1. **ECFP kernels**: Tanimoto, Dice, Otsuka, Sogenfrei, Braun-Blanquet, Faith, Forbes, Inner-Product, Intersection, Min-Max, and Rand; standard Gaussian and Laplacian kernels (directly to these ECFP representations as one-hot vectors). See Section B.1.1 for details.

2. **Pre-trained features:** SELFIESTED, SELFormer, ChemBERTa, and MLT-BERT (transformer-based with string-based inputs), GROVER (GNN-based). See Section B.1.2 for details.

3. **Global 3D features:** The Coulomb matrix (CM), bag of bonds (BOB), and SLATM equipped with Gaussian, Laplacian, and linear kernels. See Section B.1.3.

4. **Local 3D features:** local SOAP Bartók et al. (2013), FCHL19 Christensen et al. (2020), and ACSF Behler (2011) equipped with Gaussian and Laplacian kernels. See Section B.2.

**Training**  For kernel features, we use standard Kernel Ridge Regression to perform the regression task. For pre-trained features, we perform our novel *kernel probing* method: we fix a reproducing kernel $k$ defined on $\mathbb{R}^P$ and run KRR on $k$ taking the pre-trained features $\mathbf{h}_i$'s as inputs. In this paper, we choose $k$ to be Gaussian, Laplacian, or linear. Note that kernel probing with a linear kernel is equivalent to linear probing with L2-regularization.

We report the test error as the Mean Absolute Error (MAE) of the regressors trained QM9 on the right side of the Table 2, with a train size of $N_{\text{train}} = 5,000$ and test size of $N_{\text{test}} = 10,000$. Please refer to Appendix C for the results of other datasets, the ablation test on $N_{\text{train}}$ and the details of the experimental designs.

Table 2: Comparison of spectral metrics and **MAE** obtained from KRR using different molecular representations. Spectral metrics are reported for kernels with hyperparameters tuned on the $C_V$ property. The best and second-best MAE values for each property are highlighted in blue and red, respectively. The four spectral metrics quantify the richness of the kernel spectrum (direction indicated by arrows).

| Mol. Rep. | Kernel | $\alpha\downarrow$ | SSE $\uparrow$ | ID $\uparrow$ | SR $\uparrow$ | MAE $\downarrow$ | | | | | | |
| --- | --- | --- | --- | --- | --- | --- | --- | --- | --- | --- | --- | --- |
| | | | | | | Gap (eV) | $C_V$ (cal/molK) | $\Delta H$ (eV) | $U_0$ (eV) | $U_{298}$ (eV) | $G$ (eV) | ZPVE (eV) |
| ECFP6 | Tanimoto | 0.71 | 1699.71 | 13.71 | 1.30 | 0.406 | 1.499 | 415.975 | 415.982 | 415.975 | 415.992 | 0.253 |
| | Dice | 0.79 | 432.12 | 7.57 | 1.24 | 0.466 | 1.567 | 431.930 | 431.936 | 431.930 | 431.946 | 0.272 |
| | Otsuka | 0.79 | 429.57 | 7.57 | 1.24 | 0.484 | 1.648 | 458.947 | 458.954 | 458.947 | 458.964 | 0.282 |
| | Sogenfrei | **0.52** | **3121.53** | **40.52** | **2.05** | 0.381 | 1.557 | 438.923 | 438.930 | 438.923 | 438.941 | 0.254 |
| | Braun-Blanquet | 0.79 | 426.19 | 7.57 | 1.24 | 0.499 | 1.711 | 503.477 | 503.483 | 503.477 | 503.494 | 0.293 |
| | Faith | 0.83 | 1.25 | 1.02 | 1.00 | 0.481 | 1.623 | 454.478 | 454.484 | 454.478 | 454.494 | 0.283 |
| | Forbes | 0.79 | 432.12 | 7.57 | 1.24 | 0.508 | 1.695 | 473.178 | 473.185 | 473.178 | 473.196 | 0.291 |
| | Inner-Product | 0.79 | 426.19 | 7.57 | 1.24 | 0.501 | 1.728 | 496.016 | 496.022 | 496.016 | 496.032 | 0.293 |
| | Intersection | 0.83 | 1.13 | 1.01 | 1.00 | 0.488 | 1.656 | 465.531 | 465.537 | 465.531 | 465.547 | 0.287 |
| | Min-Max | 0.71 | 1699.71 | 13.71 | 1.30 | 0.406 | 1.499 | 415.975 | 415.982 | 415.975 | 415.992 | 0.253 |
| | Rand | 0.83 | 1.13 | 1.01 | 1.00 | 0.481 | 1.623 | 454.477 | 454.483 | 454.477 | 454.493 | 0.283 |
| | Gaussian | 0.91 | 1.00 | 1.00 | 1.00 | 0.496 | 1.653 | 458.080 | 458.086 | 458.818 | 459.986 | 0.291 |
| | Laplacian | 0.89 | 1.00 | 1.00 | 1.00 | 0.466 | 1.626 | 454.330 | 454.336 | 454.330 | 454.345 | 0.285 |
| SELFIESTED | Gaussian | 4.79 | 1.00 | 1.00 | 1.00 | 0.439 | 0.466 | 78.007 | 78.011 | 78.005 | 78.012 | 0.054 |
| | Laplacian | **0.85** | 4.73 | 1.21 | 1.00 | 0.439 | 0.576 | 131.868 | 131.869 | 131.868 | 131.872 | 0.087 |
| | Linear | 1.96 | 4.95 | 1.39 | 1.01 | 0.462 | 0.667 | 159.911 | 159.927 | 159.917 | 159.937 | 0.095 |
| SELFormer | Gaussian | 2.92 | 1.05 | 1.01 | 1.00 | 0.604 | 1.405 | 512.099 | 512.106 | 512.099 | 512.118 | 0.235 |
| | Laplacian | 0.91 | 3.34 | 1.12 | 1.00 | 0.659 | 1.671 | 553.634 | 553.641 | 553.634 | 553.653 | 0.303 |
| | Linear | 8.41 | 2.14 | 1.13 | 1.00 | 0.639 | 1.592 | 552.926 | 553.118 | 552.979 | 552.983 | 0.275 |
| MLT-BERT | Gaussian | 4.06 | 1.00 | 1.00 | 1.00 | 0.507 | 0.977 | 213.632 | 213.636 | 213.632 | 213.642 | 0.173 |
| | Laplacian | 1.06 | 4.84 | 1.27 | 1.00 | 0.609 | 1.350 | 328.146 | 328.151 | 328.146 | 328.161 | 0.246 |
| | Linear | 11.50 | 1.82 | 1.12 | 1.00 | 0.579 | 1.294 | 307.626 | 307.630 | 307.626 | 307.640 | 0.240 |
| ChemBERTa | Gaussian | 4.74 | 1.00 | 1.00 | 1.00 | 0.466 | 1.701 | 488.439 | 488.445 | 488.439 | 488.455 | 0.297 |
| | Laplacian | 0.95 | 1.03 | 1.00 | 1.00 | 0.483 | 1.739 | 505.729 | 505.735 | 505.729 | 505.747 | 0.326 |
| | Linear | 7.11 | **8.58** | **1.76** | **1.05** | 0.526 | 1.866 | 539.814 | 539.982 | 539.717 | 540.051 | 0.325 |
| GROVER$_{base}$ | Gaussian | 3.15 | 1.00 | 1.00 | 1.00 | 0.363 | 0.996 | 261.153 | 261.169 | 261.168 | 261.187 | 0.113 |
| | Laplacian | 0.89 | 4.27 | 1.20 | 1.00 | 0.369 | 0.991 | 292.025 | 292.031 | 292.025 | 292.040 | 0.128 |
| | Linear | 1.93 | 2.84 | 1.21 | 1.00 | 0.372 | 1.017 | 299.277 | 299.283 | 299.277 | 299.294 | 0.130 |
| GROVER$_{large}$ | Gaussian | 1.98 | 1.00 | 1.00 | 1.00 | 0.354 | 0.934 | 271.643 | 271.649 | 271.643 | 271.658 | 0.119 |
| | Laplacian | 0.88 | 5.66 | 1.26 | 1.00 | 0.358 | 0.932 | 273.771 | 273.777 | 273.771 | 273.786 | 0.120 |
| | Linear | 1.84 | 2.97 | 1.22 | 1.01 | 0.361 | 0.931 | 272.434 | 272.439 | 272.434 | 272.449 | 0.119 |
| CM | Gaussian | 1.72 | 4.91 | 1.32 | 1.00 | 0.662 | 0.558 | 0.814 | 0.814 | 0.814 | 0.813 | 0.023 |
| | Laplacian | 1.54 | 1.58 | 1.05 | 1.00 | 0.464 | 0.341 | 4.110 | 4.110 | 4.110 | 4.110 | 0.013 |
| | Linear | 9.24 | 1.75 | 1.08 | 1.00 | 0.772 | 0.973 | 2.639 | 2.639 | 2.639 | 2.646 | 0.040 |
| BOB | Gaussian | 3.13 | **7.24** | **1.81** | **1.10** | 0.456 | 0.518 | 0.755 | 0.753 | 0.755 | 0.754 | 0.018 |
| | Laplacian | 1.54 | 1.65 | 1.08 | 1.00 | 0.312 | 0.207 | 4.476 | 4.476 | 4.476 | 4.477 | 0.009 |
| | Linear | 10.84 | 3.20 | 1.36 | 1.03 | 0.685 | 0.770 | 0.835 | 0.833 | 0.835 | 0.830 | 0.025 |
| SLATM | Gaussian | 3.64 | 1.00 | 1.00 | 1.00 | 0.220 | 0.093 | 2.509 | 2.508 | 2.508 | 2.508 | 0.004 |
| | Laplacian | **1.35** | 1.18 | 1.02 | 1.00 | 0.229 | 0.140 | 22.340 | 22.340 | 22.340 | 22.341 | 0.009 |
| | Linear | 4.43 | 1.79 | 1.13 | 1.00 | 0.403 | 0.178 | 3.971 | 3.971 | 3.971 | 3.970 | 0.004 |
| SOAP | Gaussian | 4.50 | 1.18 | 1.03 | 1.00 | 0.289 | 0.087 | 0.063 | 0.063 | 0.063 | 0.063 | 0.003 |
| | Laplacian | 1.30 | 1.53 | 1.07 | 1.00 | 0.332 | 0.142 | 0.137 | 0.137 | 0.137 | 0.136 | 0.005 |
| FCHL19 | Gaussian | 3.13 | 1.21 | 1.03 | 1.00 | 0.271 | 0.081 | 0.055 | 0.055 | 0.055 | 0.055 | 0.003 |
| | Laplacian | **1.29** | 1.78 | 1.10 | 1.00 | 0.288 | 0.116 | 0.138 | 0.137 | 0.138 | 0.137 | 0.004 |
| ACSF | Gaussian | 5.39 | 1.19 | 1.03 | 1.00 | 0.329 | 0.149 | 0.162 | 0.162 | 0.162 | 0.161 | 0.005 |
| | Laplacian | 1.42 | **2.53** | **1.19** | **1.00** | 0.372 | 0.187 | 0.230 | 0.230 | 0.230 | 0.228 | 0.006 |

## 3.2 Correlation between Feature Spectrum and Performance

**Spectral Metrics** For each type of molecular feature, we compute the empirical eigenspectrum $\mu_1, \ldots, \mu_n$ of its Gram matrix $\boldsymbol{K} = \frac{1}{N}\mathbf{H}^\top\mathbf{H}$ or $\frac{1}{N}\boldsymbol{\Phi}^\top\boldsymbol{\Phi}$ in $\mathbb{R}^{N\times N}$. Then we evaluate 4 spectral metrics to quantify its richness: polynomial decay rate ($\alpha\downarrow$), spectral Shannon entropy (SSE $\uparrow$), intrinsic dimension (ID $\uparrow$), and stable rank (SR $\uparrow$). The arrows indicate the direction corresponding to richer spectral features, with formal definitions provided in Section D.

In our experiments, we find that most kernel spectra are dominated by a single or a few large leading eigenvalues (e.g., $\mu_1$), followed by a sharply decaying tail (see Fig. 2 and figures in Section C.1 for QM9). Motivated by this structure, SSE, ID, and SR are computed from the full spectrum, while the power-law exponent $\alpha$ is estimated by fitting $\log\mu_i$ vs. $\log i$ over the top 50% of eigenvalues, excluding the noise-dominated tail.

The spectral metrics are reported in the middle of Tables 2 for the QM9 dataset. Results for the ESOL, FreeSolv, and Lipophilicity datasets are presented separately in Appendix C.

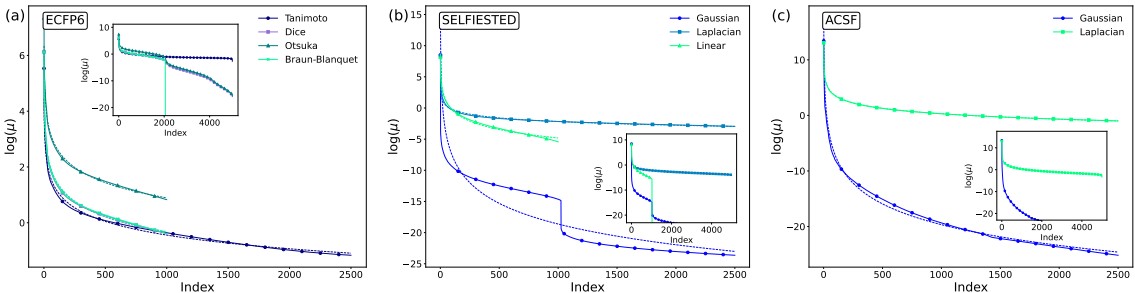

Figure 2: Kernel eigenvalue spectra with insets highlighting that nearly half of the eigenvalues are close to zero (main plots) for different molecular representations. Results are shown for (a) ECFPs, (b) SELFIESTED, and (c) local 3D descriptor-based kernels.

**Correlation Analysis** To investigate whether spectral richness translates to improved predictive accuracy, we evaluate the relationships between our four spectral metrics and the average $R^2$ score across the seven QM9 molecular properties (visualized in Fig. 3). These relationships are quantified using Pearson correlation coefficients ($\hat{r}$) along with their 95% confidence intervals, as detailed in Table 3. Because a smaller power-law decay parameter $\alpha$ signifies a richer spectrum, we report the correlation with $-\alpha$ so that a positive $\hat{r}$ consistently indicates that greater spectral richness aligns with better predictive performance.

Our empirical findings reveal that the common self-supervised learning heuristic—"richer spectra yield better performance"—fails to hold universally. Across the board, correlations are generally weak, frequently statistically insignificant, and many confidence intervals overlap zero. However, distinct behaviors emerge when analyzing specific representation types:

- **ECFP kernels:** Correlations are generally moderate and positive. However, only the spectral Shannon entropy (SSE) achieves statistical significance, while the remaining metrics yield confidence intervals spanning zero.
- **Pre-trained features:** Transformer-based kernels exhibit mixed, statistically inconclusive behavior. The decay rate ($-\alpha$) displays a weak positive correlation, whereas SSE, intrinsic dimension (ID), and stable rank (SR) show weak negative trends.
- **Global 3D features:** These present a notable divergence. The power-law decay rate ($-\alpha$) exhibits a strong, statistically significant positive correlation with performance, whereas SSE, ID, and SR remain weakly negative and non-significant.
- **Local 3D features:** In direct contrast to the SSL heuristic, local 3D kernels show consistently negative correlations across SSE, ID, and SR. This suggests that increased spectral richness does not inherently improve accuracy and may even prove detrimental, though wide confidence intervals leave these specific trends statistically inconclusive.

In summary, **spectral richness alone is an unreliable predictor of downstream performance**; its practical impact hinges critically on the underlying nature of the molecular representation.

### 3.3 Ablation Study on Feature Spectrum

To understand how specific features shape the whole feature spectrum, we conducted a comprehensive feature ablation study across four kernel families (Tanimoto, Dice, Laplacian, and Gaussian) using three representations: ECFP, BOB, and SELFIESTED. For the sparse, rule-based ECFP and BOB representations, we implemented a frequency-weighted ablation scheme, removing $\tilde{P}$ features with a probability proportional to their occurrence dataset-wide. This ensures that the perturbations disrupt frequent, chemically meaningful molecular fragments. For the dense, non-interpretable SELFIESTED embeddings, we uniformly and randomly removed embedding dimensions as a comparative baseline.

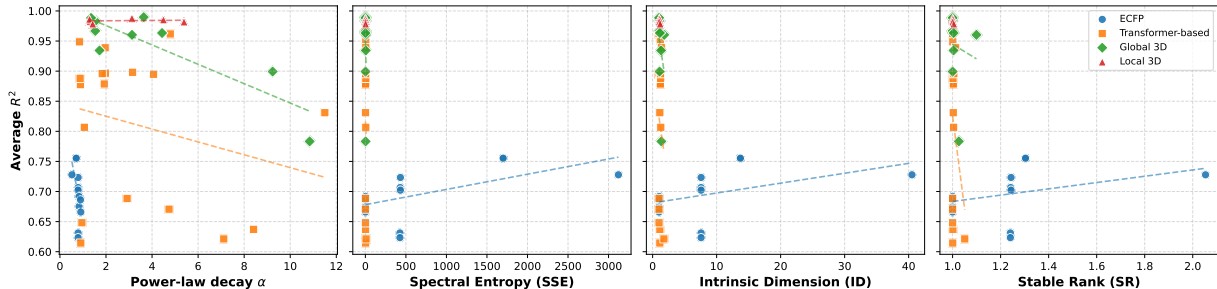

Figure 3: Correlation between spectral metrics and average $\mathbf{R}^2$ across molecular kernel categories. Dotted lines indicate the best-fit linear trend for each category.

Table 3: Pearson correlation coefficients ($\hat{r}$) with 95% confidence intervals (CI) among the spectral metrics $-\alpha$, SSE, ID, and SR, which align with the notion of spectral richness, and average $\mathbf{R}^2$ across molecular kernel categories. Correlations whose 95% CI excludes zero are shown in **bold**.

| Mol. Rep. | $-\alpha \uparrow$ | SSE $\uparrow$ | ID $\uparrow$ | SR $\uparrow$ |
|---|---|---|---|---|
| ECFP | 0.501 | **0.585** | 0.430 | 0.362 |
| | [-0.069, 0.825] | [0.050, 0.859] | [-0.158, 0.793] | [-0.236, 0.761] |
| Transformer-based | 0.256 | -0.047 | -0.113 | -0.290 |
| | [-0.239, 0.646] | [-0.503, 0.429] | [-0.551, 0.374] | [-0.667, 0.204] |
| Global 3D | **0.863** | -0.195 | -0.281 | -0.121 |
| | [0.465, 0.971] | [-0.761, 0.539] | [-0.797, 0.471] | [-0.727, 0.591] |
| Local 3D | -0.124 | -0.646 | -0.655 | -0.716 |
| | [-0.850, 0.764] | [-0.956, 0.348] | [-0.958, 0.334] | [-0.966, 0.228] |

As illustrated in Figs. 4 and 5, the structural response of the eigenvalue spectrum to ablation varies significantly by representation kind. ECFP-based Tanimoto and Dice kernels exhibit remarkable robustness to feature loss, maintaining stable spectra even when a quarter of the features ($\tilde{P} = 512$) are removed. Continuous kernels (Gaussian and Laplacian) built on ECFP and BOB are highly sensitive to the length-scale parameter $\ell$—where larger values accelerate eigenvalue decay—but show localized spectral shifts under ablation. Meanwhile, the dense SELFIESTED embeddings prove to be the most robust; their leading eigenvalues remain stable and their spectra decay smoothly even after removing hundreds of dimensions, confirming that optimized latent spaces successfully distribute chemical information across highly correlated dimensions.

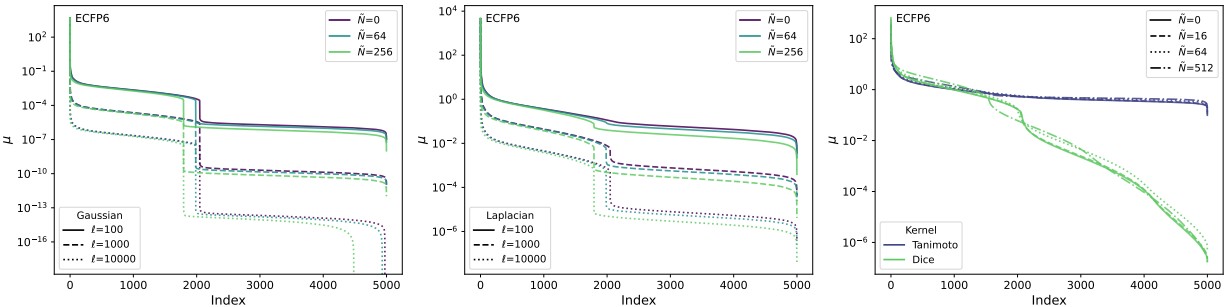

Figure 4: Kernel eigenvalue spectra for (a) Gaussian, (b) Laplace, and (c) Tanimoto and Dice kernels using ECFPs across various ablation levels ($\tilde{P}$) and length-scales ($\ell$). Higher $\tilde{P}$ contracts the spectra, signaling a reduced effective dimensionality.

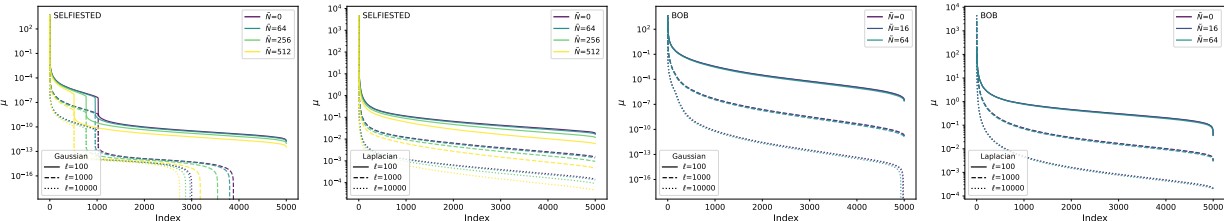

Figure 5: Kernel eigenvalue spectra for (a–b) SELFIESTED and (c–d) BOB representations using Laplace and Gaussian kernels under feature ablation ($\tilde{P}$) and varied length-scales ($\ell$).

Crucially, our spectral metrics serve as robust and informative indicators of how these ablations reshape the underlying kernel space (Tables 6 and 7). For ECFP-based kernels, SSE, ID, and SR increase sharply with $\tilde{P}$, mirroring a near-twofold increase in downstream MAE across all properties. This synchronous shift confirms that the metrics successfully capture the degradation of essential predictive features.

Conversely, for continuous Gaussian and Laplacian kernels across all representations, SSE, ID, and SR remain close to 1 regardless of $\tilde{P}$, whereas the power-law decay exponent $\alpha$ uniquely captures the smoothing effect of the length scale $\ell$. Most notably, for the BOB representation paired with a Laplacian kernel ($\ell = 100$), increasing ablation ($\tilde{P}$) actually reduces SSE, ID, and SR while simultaneously *improving* downstream MAE. This exceptional case highlights the diagnostic power of our metrics: they can successfully detect when ablation prunes redundant noise from a kernel spectrum rather than stripping away useful representation capacity.

### 3.4 Truncated Kernel Ridge Regression

Our findings show that the required truncation threshold ($r/N_{\text{train}}$) varies substantially across representation types. For global 3D (CM, BOB, SLATM) and local 3D (SOAP, FCHL19, ACSF) representations, retaining fewer than 1% of the eigenvalues is sufficient to recover 95% of the performance for energy-related properties ($U_0$, $U_{298}$, $\Delta H$, and $G$), while fewer than 10% suffices for $C_V$ and $ZPVE$; see Fig. 6. These results indicate that the leading eigenvalues capture nearly all of the informative spectral content for these representations. Transformer-based representations require a moderately larger fraction of eigenvalues (3–28%). In contrast, fingerprint-based kernels show the largest variability: some (e.g., Braun–Blanquet, Forbes, Inner Product) achieve 95% recovery with fewer than 3% of eigenvalues, while others (e.g., Tanimoto, Sogenfrei, Min–Max) require nearly the full spectrum for certain properties, indicating a less compact spectral structure; see Fig. 6.

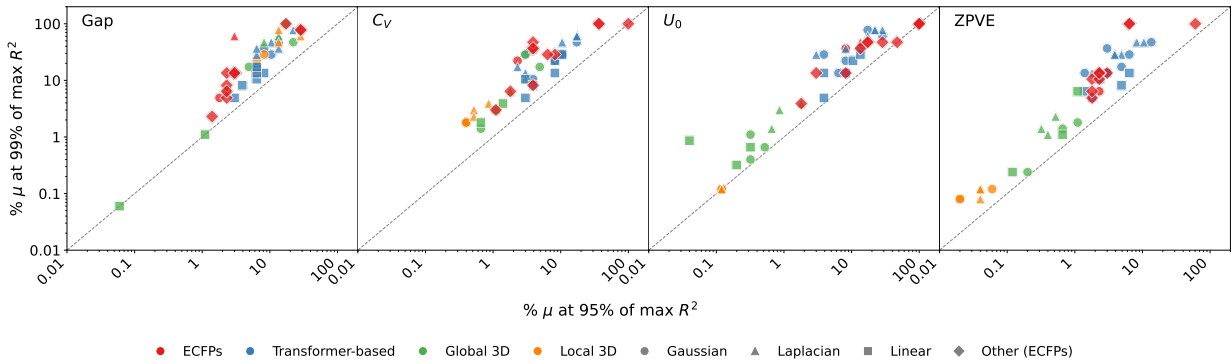

Figure 6: Eigenvalue truncation thresholds to reach 95% and 99% of maximum $R^2$.

# 4 Discussion

In this section, we discuss the main implications of our empirical findings for (i) kernel theory and self-supervised learning, (ii) practical molecular-chemistry practice, and (iii) limitations and avenues for future work.

## 4.1 Insights for Kernel Theory and Self-Supervised Learning

**Fingerprint and Transformer Kernels** ECFP kernels are the only category showing consistently positive correlations between spectral richness and predictive performance, whereas transformer-based kernels show mixed behavior. Within the fingerprint family, ECFP6 consistently yields richer spectra than ECFP4 across all kernels and correspondingly achieves lower MAEs, with the Gap property being a notable exception, where ECFP4 slightly outperforms ECFP6. Surprisingly, the Sogenfrei kernel has the best spectral metrics values and the second-lowest MAEs, except for Gap. Contrary to the Tanimoto kernel, the preferred kernel in cheminformatics, outperforms all other ECFP-based kernels and has the second-highest spectral metrics. In this narrow setting, the common SSL heuristic—"richer spectra yield better performance"—appears to hold. Our spectral analysis provides a principled explanation: the key difference lies in the richness of the spectral tail, with Tanimoto retaining more information in the lower-ranked eigenvectors (see Fig. 7). This might suggest that ECFP-based kernels, being hand-designed, may be fundamentally different from pretrained-derived features: they already encode domain knowledge in the representation itself, so most relevant information is concentrated in the top eigenvectors, making spectral richness less decisive.

Transformer-based kernels, where representations are generated from models pretrained on large chemical corpora and then evaluated on unseen QM9 tasks in a setup analogous to SSL, provide a weak and inconsistent support for the heuristic that greater spectral improves predictive performance. Our results show that only the spectral decay coefficient ($-\alpha$) has a weak positive correlation, while SSE, ID and SR all show weak negative correlations. Moreover, the confidence intervals for all four coefficients span zero, indicating no statistically significant relationship. Consistent with this observation, the kernels with the lowest MAE does not come from representations with high spectral richness (see Table 2). Furthermore, among the transformer-based models, SELFormer shows the lowest spectral metric values, while ChemBERTa achieves the highest spectral richness.

**3D Kernels** For 3D global and local kernels, the evidence indicates that both kernel families exhibit predominantly negative correlations. Only in the case of global kernels does $-\alpha$ show a positive correlation. As shown in Table 2, FCHL19 outperforms SOAP and ASCF in terms of MAE; however, it does not correspond to the kernel with the highest spectral metric values. Instead, ACSF attains the largest spectral metric values, further challenging a consistent relationship between spectral richness and predictive accuracy. Within global 3D kernels, BOB with a Gaussian kernel shows the largest SSE, ID and SR values and achieves the best performance for the energy-related properties, whereas SLATM with a Gaussian kernel attains the highest accuracy on Gap, $C_V$, and ZPVE.

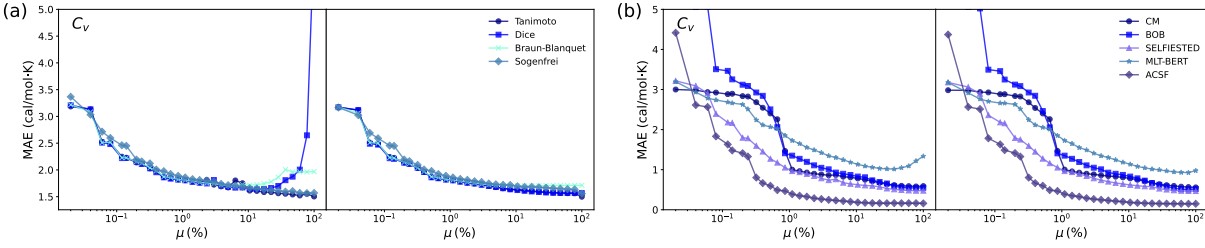

Figure 7: MAE for the heat capacity ($C_V$) property as a function of truncation level ($\mu(\%)$ for (a) selected ECFP-based kernels and (b) four various global (CM, BOB, SELFIESTED, MLT-BERT) kernels and a single local (ACSF) kernel, all with a Gaussian kernel. Left sub-panel: without regularization; right sub-panel: with regularization.

**Tikhonov Regularization versus Truncation** Tikhonov regularization has long been a standard technique in ML to mitigate overfitting to label noise. In kernel methods, it works by penalizing the use of high-frequency eigenfunctions in fitting the data. On the other hand, in a linear regression setting Hansen (1987) showed that truncation achieves a similar effect by explicitly discarding the tail of the spectrum, thereby removing high-frequency components from the hypothesis space. However, to the best of our knowledge, our paper is the first to discover the same effect in molecular kernels. As shown in Fig. 7, the performance of the best ridgeless truncated KRR (left panel) is comparable to that of the fully regularized KRR (right panel). This observation provides a possible explanation for why richer spectra may sometimes harm generalization: additional eigenfunctions in the tail can facilitate overfitting rather than improve predictive accuracy, and any regularization to avoid overfitting would harm the accuracy. Notably, this phenomenon is not unique to ECFP-based kernels, but is also observed across other kernel categories (see Section C.2 for additional plots).

## 4.2 Insights for Molecular Chemistry

**First Comprehensive Results** Pretrained molecular embedding models have recently attracted significant interest in chemistry, particularly for small molecules, as they are increasingly adopted for supervised learning tasks. Related work has applied pretrained embeddings in a kernel framework for proteins Griffiths et al. (2023); Groth et al. (2024). However, these efforts were limited to kernel construction without further spectral analysis, such as ours. In contrast, this work is the first to explore a kernel-based framework built upon pretrained molecular embedding models for chemistry while also analyzing their spectral characteristics.

**Transformer-based Representations** Previous work has mainly applied linear or MLP-based regression to transformer-derived molecular representations Praski et al. (2025), motivated by the high dimensionality of embeddings, where kernel matrices often resemble their linearization—a weighted sum of the covariance matrix, identity, and a rank-one term El Karoui (2010). In contrast, we show that KRR with a Gaussian kernel outperforms the linear baseline on QM9 dataset, indicating that higher-order terms capture additional information beyond linear covariance. Notably, GROVER$_{large}$ with a Gaussian kernel achieves the best performance on the Gap property, whereas SELFIESTED performs best on the other properties. This suggests an alternative way to evaluate SSL models—via spectral metrics derived from their kernel matrices—which we leave for future work.

**MoleculeNet Benchmarks** In their benchmark Wu et al. (2018), KRR with ECFP4 and a Gaussian kernel achieved an RMSE of $\sim 0.9$. Using a broader set of representations and kernels (Table 5), we find that GROVER$_{large}$ with a Gaussian kernel achieves the best performance on Lipophilicity, while SELFIESTED combined with a Gaussian or Laplacian kernel performs best on FreeSolv and ESOL, respectively. Among fingerprint-based kernels, ECFP4 with the Tanimoto kernel consistently yields the strongest performance across all three datasets. Overall, several of our kernel-based models outperform the MoleculeNet KRR baseline and remain competitive with their neural models, with results reported in terms of RMSE (GCNNWu et al. (2018): 0.67±0.04; LSTM-attention: 0.60±0.04).

As in QM9, no single kernel type consistently dominates across transformer-based representations on the MoleculeNet benchmarks: Gaussian kernels perform better for GROVER and SELFIESTED, while Laplacian kernels are slightly better for SELFormer, MLT-BERT, and ChemBERTa. However, in neither case do the spectral metrics exhibit a clear correlation with predictive performance. Across Lipophilicity, ESOL, and FreeSolv, ECFP6 with Tanimoto consistently shows the richest and slowest-decaying eigenvalue spectrum (Table 5), yet does not achieve the best predictive accuracy. Conversely, transformer-based representations such as GROVER and SELFIESTED with faster spectral decay lead to the best KRR performance on these benchmarks. Thus, our results confirm the same trend seen in QM9: richer spectral structure does not imply better kernel regression performance, and alignment between representation geometry and the kernel function is more important than embedding capacity.

**3D Descriptors** The comparison between global and local kernels, whose representations are built on Cartesian coordinates, has sparked the latter development in molecular kernels Thant et al. (2025). However, we found that global 3D kernels are more susceptible to drastic effects in their accuracy when hyperparameter search is found to be suboptimal, contrary to local 3D kernels. For global 3D representations on QM9 dataset, SLATM consistently outperformed the other descriptors on Gap, $C_V$, and ZPVE regardless of kernel choice. In contrast, for the energy-related properties, CM achieved the best performance.

### 4.3 Limitations and Future Work

Despite our systematic experiments and analyses, several limitations remain.

**Diagnostics for Molecular Features** While this paper provides evidence that the feature heuristics predominantly used in SSL image task does not work in molecular setting, a plausible alternative is still missing and unexplored.

**Learnable Backbones** Although we assume the pre-trained features are frozen in this paper, analogous to the fixed kernel features, practitioners often fine-tune the pre-trained features in the downstream task as well as the prediction head. How the feature spectrum evolves also shed lights into the mechanism of modern machine learning.

**Representations and Kernels** We did not include recent hybrid graph-based encoders like Mol2Vec Jaeger et al. (2018), which may reveal distinct spectral behaviors. Similarly, quantum-inspired kernels derived from molecular graph circuits Schuld et al. (2020); Torabian & Krems (2025) represent another promising direction for applying our framework to evaluate the structure and capacity of emerging methods in chemistry and materials science.

## 5 Conclusion

We presented the first systematic spectral analysis of molecular features for property prediction on QM9 and 3 MoleculeNet benchmarks, spanning kernels with ECFP, pretrained transformer-based features, and global or local 3D descriptors as inputs. Our results show that spectral richness is not a universal predictor of performance. Pearson correlations reveal a consistent pattern only for ECFP-based kernels, where all four spectral metrics are positively correlated with accuracy. In contrast, transformer-based and global 3D kernels show mixed behavior, with $-\alpha$ weakly positive but SSE, ID, and SR negative, while local 3D kernels reverse the heuristic entirely, with all metrics negatively correlated. The truncation analysis supports this mixed behavior. For local 3D features and thermodynamic QM9 targets, fewer than 2% of eigenvalues are often sufficient to recover 95% of the maximum performance, reaching as low as 0.02% in some cases. By contrast, ECFPs and transformer-based features for properties such as the HOMO–LUMO gap typically require substantially more eigenvalues, in some cases nearly the full spectrum. Overall, these findings call into question the common heuristic that "richer features yield better generalization." More broadly, our study offers practical guidance for pairing molecular representations with kernels and opens a new avenue for bridging spectral analysis between self-supervised learning and kernel methods.

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

# A  Appendix

The appendix is organized as follows. Section B introduces the molecular representations and kernel functions considered in this work, including fingerprint-based kernels, pretrained text-embedding models, graph-based and Cartesian coordinate–based representations. As noted in the main text, we use the term **3D** kernels to refer to kernels derived from Cartesian coordinates, whether global or local. Section C presents supplementary experimental results. Section D provides detailed definitions of the four spectral metrics. Finally, Section E contains proofs omitted from the main text.

# B  Molecular Kernels

Here, we briefly summarize molecular kernels that are based on molecular representations, which can broadly be divided into two categories:

**Definition 1** (Global Molecular Representation). *Let $\mathcal{M}$ denote a molecule and $\phi : \mathcal{M} \to \mathbb{R}^d$ be a function that maps a molecule to a d-dimensional vector of descriptors that summarize the entire structure (e.g., fingerprints, Coulomb matrix eigenvalues, or learned embeddings by encoding models).*

**Definition 2** (Local Molecular Representation). *Let $\mathcal{M}$ denote a molecule composed of Na atoms, where each atom is represented by $z_\ell$ containing Cartesian coordinates and nuclear information such as atomic number. A local representation is given by a function $\phi_\ell : z_\ell \mapsto \mathbb{R}^d$ that encodes atomic environments based on the arrangement of neighboring atoms. Examples include the Smooth Overlap of Atomic Positions and many-body distribution functions.*

Due to the existence of $\phi$ and $\phi_\ell$, there are two main families of molecular kernels: global and local molecular kernels.

**Definition 3** (Global Molecular Kernel). *A* global molecular kernel *is a positive-definite function $k_{\text{global}} : \mathcal{M}_i \times \mathcal{M}_j \to \mathbb{R}$ defined as*

$$k_{global}(\mathcal{M}_i, \mathcal{M}_j) = \kappa\big(\phi(\mathcal{M}_i), \phi(\mathcal{M}_j)\big), \tag{2}$$

*where $\kappa : \mathbb{R}^d \times \mathbb{R}^d \to \mathbb{R}$ is a positive-definite kernel function comparing global descriptor vectors computed with $\phi$.*

A prominent example of a global kernel is obtained when $\phi$ is computed via extended connectivity fingerprints (ECFPs) Rogers & Hahn (2010). ECFPs are fixed-length hashed descriptors generated by iteratively encoding atom-centered circular neighborhoods (the Morgan algorithm) up to a predefined radius $r$. The resulting binary vector, $z_i^\top = \phi_{\text{ECFP}}(\mathcal{M}_i)^\top = [1, 0, 1, \cdots, 1]^\top$, captures the 2D molecular topology (and, optionally, chirality) in a global form. When using $\phi_{\text{ECFP}}(\mathcal{M})$ as the descriptor, similarity can be quantified through fingerprint-specific kernels such as

$$k_{\text{Tanimoto}}(\mathcal{M}_i, \mathcal{M}_j) = \sigma_f^2 \cdot \frac{\langle x_i, x_j \rangle}{|x_i|_2^2 + |x_j|_2^2 - \langle x_i, x_j \rangle}, \quad k_{\text{Dice}}(\mathcal{M}_i, \mathcal{M}_j) = \sigma_f^2 \cdot \frac{2\langle x_i, x_j \rangle}{|x_i|_1 + |x_j|_1}, \tag{3}$$

where $\sigma_f$ is a kernel hyperparameter, $x = \phi_{\text{ECFP}}(\mathcal{M})$, $\langle x_i, x_j \rangle = x_i^\top x_j$, and $|x_j|_p$ is the $p$-norm of the fingerprint vector. We present other ECFP-based kernels in Section B.1.1.

Another widely used class of global descriptors arises from *data-driven molecular embeddings*, where $\phi$ is learned from large corpora of molecular strings such as SMILES or SELFIES. Examples include models such as `SELFIESTED`, `SELFormer`, and `MLT-BERT`, which leverage transformer-based language models to capture chemical semantics. Unlike discrete fingerprints, these embeddings yield continuous-valued feature vectors, enabling the use of standard isotropic kernels such as Gaussian, Laplacian, or linear. Additional details of transformer-based global representations are provided in Section B.1.2.

Beyond data-driven embeddings, global representations can also incorporate explicit geometrical information. A classical example is the Coulomb Matrix (CM) Rupp et al. (2012), which encodes pairwise Coulombic interactions between atoms. Other notable global descriptors include the bag of bonds (BoB) Hansen et al. (2015) and the spectrum of London and Axilrod–Teller–Muto (SLATM) Huang & von Lilienfeld (2020). BoB, inspired by the bag-of-words algorithm in natural language processing, extends the CM by grouping pairwise interactions into "bags" according to bond type. SLATM, in contrast, is based on many-body expansions: it represents molecular structures by approximating atomic charge densities with Gaussian functions scaled by interatomic potentials. Additional details of global kernels are provided in Section B.1.

**Definition 4** (Local Molecular Kernel). *A* local molecular kernel *is a positive-definite function of two molecules, defined as*

$$k_{local}(\mathcal{M}_i, \mathcal{M}_j) = \sum_{\ell_i=1}^{Na_i} \sum_{\ell_j=1}^{Na_j} g(Z_{\ell_i}, Z_{\ell_j}) \, \kappa\big(\phi_\ell(\boldsymbol{z}_{\ell_i}), \phi_\ell(\boldsymbol{z}_{\ell_j})\big), \tag{4}$$

*where $\boldsymbol{z}_{\ell_i}$ denotes the position and chemical identity of the $\ell_i$-th atom in $\mathcal{M}_i$, $\phi_\ell$ maps its local chemical environment to a descriptor (e.g., SOAP, FCHL19, ACSF), and $\kappa$ is a positive-definite kernel function (such as Gaussian or Laplacian) that measures similarity between atomic environments. The function $g(Z_{\ell_i}, Z_{\ell_j})$ compares atomic species, typically defined as a Kronecker delta on the atomic numbers, i.e. $g(Z_{\ell_i}, Z_{\ell_j}) = \delta(Z_{\ell_i} = Z_{\ell_j})$.*

Although global descriptors capture holistic molecular information, they may struggle to generalize across molecules with different sizes or conformations. Much of the recent work on molecular kernel development has therefore focused on incorporating geometric information at the atomic level. *Local kernels* address this by encoding atomic environments within a cutoff radius, making them naturally suited to enforce invariances (e.g., translation, rotation, and permutation) and improving transferability across chemical space. Prominent examples include the Smooth Overlap of Atomic Positions (SOAP) Bartók et al. (2013), Faber–Christensen–Huang–Lilienfeld (FCHL) Faber et al. (2018); Christensen et al. (2020), and many-body distribution functions (MBDF) Khan et al. (2023); Khan & von Lilienfeld (2024). Additional details of local kernels are provided in Section B.2.

## B.1 Global Molecular Kernels/Representations

### B.1.1 Extended-Connectivity Fingerprints

One of the most common global molecular representations is the extended-connectivity fingerprints (ECFPs) Rogers & Hahn (2010). Here is a list of some of the global molecular kernels based on the ECFP representation,

$$k_{\text{Braun-Blanquet}} = \frac{\langle \boldsymbol{x}_1, \boldsymbol{x}_2 \rangle}{\max(|\boldsymbol{x}_1|, |\boldsymbol{x}_2|)}, \qquad k_{\text{Dice}} = \frac{2\langle \boldsymbol{x}_1, \boldsymbol{x}_2 \rangle}{|\boldsymbol{x}_1| + |\boldsymbol{x}_2|} \tag{5}$$

$$k_{\text{Faith}} = \frac{2\langle \boldsymbol{x}_1, \boldsymbol{x}_2 \rangle + d_0}{2d}, \qquad k_{\text{Forbes}} = \frac{d\langle \boldsymbol{x}_1, \boldsymbol{x}_2 \rangle}{|\boldsymbol{x}_1| + |\boldsymbol{x}_2|} \tag{6}$$

$$k_{\text{Inner-Product}} = \langle \boldsymbol{x}_1, \boldsymbol{x}_2 \rangle = \boldsymbol{x}_1^\top \boldsymbol{x}_2, \qquad k_{\text{Intersection}} = \langle \boldsymbol{x}_1, \boldsymbol{x}_2 \rangle + \langle \boldsymbol{x}_1', \boldsymbol{x}_2' \rangle \tag{7}$$

$$k_{\text{MinMax}} = \frac{|\boldsymbol{x}_1| + |\boldsymbol{x}_2| - |\boldsymbol{x}_1 - \boldsymbol{x}_2|}{|\boldsymbol{x}_1| + |\boldsymbol{x}_2| + |\boldsymbol{x}_1 - \boldsymbol{x}_2|}, \qquad k_{\text{Otsuka}} = \frac{\langle \boldsymbol{x}_1, \boldsymbol{x}_2 \rangle}{\sqrt{|\boldsymbol{x}_1| |\boldsymbol{x}_2|}} \tag{8}$$

$$k_{\text{Rogers-Tanimoto}} = \langle \boldsymbol{x}_1, \boldsymbol{x}_2 \rangle + \frac{d_0}{2|\boldsymbol{x}_1|} + 2|\boldsymbol{x}_2| - 3\langle \boldsymbol{x}_1, \boldsymbol{x}_2 \rangle + d_0, \qquad k_{\text{Rand}} = \frac{\langle \boldsymbol{x}_1, \boldsymbol{x}_2 \rangle + d}{n} \tag{9}$$

$$k_{\text{Russel-Roa}} = \frac{\langle \boldsymbol{x}_1, \boldsymbol{x}_2 \rangle}{n}, \qquad k_{\text{Sogenfei}} = \frac{\langle \boldsymbol{x}_1, \boldsymbol{x}_2 \rangle^2}{|\boldsymbol{x}_1| + |\boldsymbol{x}_2|} \tag{10}$$

$$k_{\text{Soakl-Sneath}} = \frac{\langle \boldsymbol{x}_1, \boldsymbol{x}_2 \rangle}{2|\boldsymbol{x}_1|} + 2|\boldsymbol{x}_2| - 3\langle \boldsymbol{x}_1, \boldsymbol{x}_2 \rangle, \qquad k_{\text{Tanimoto}} = \frac{\langle \boldsymbol{x}_1, \boldsymbol{x}_2 \rangle}{\|\boldsymbol{x}_1\|^2 + \|\boldsymbol{x}_2\|^2 - \langle \boldsymbol{x}_1, \boldsymbol{x}_2 \rangle} \tag{11}$$

where:

- $\boldsymbol{x}_i$ is the global representation of the molecule using the ECFPs; $\boldsymbol{x}_i = \phi_{\text{ECFP}}(\mathcal{M}_i)$, for example, $\boldsymbol{x}_i^\top = [1, 0, 1, \cdots, 1]^\top$.

- $\langle \boldsymbol{x}_i, \boldsymbol{x}_j \rangle$ denotes the inner product.

- $\boldsymbol{x}_i'$ is the bit-flipped vector of $\boldsymbol{x}_i$.

- $|\boldsymbol{x}_i|$ represents the $L_1$ norms of $\boldsymbol{x}_i$.

- $d_0$ is the number of common zeros, and $d$ is the dimension of the input vectors,

### B.1.2 Pretrained Molecular Embedding Models

Pretrained molecular embedding models Praski et al. (2025) have become a standard approach for molecular property prediction. These models are trained on large molecular corpora to produce embedding vectors $\boldsymbol{z} \in \mathbb{R}^d$, which can then be used for downstream regression tasks. We briefly describe the five pretrained transformer-based models used in our analysis:

- `SELFIESTED`: a BART-based encoder–decoder model for SELFIES, with 358M parameters, 12 layers, and 16 attention heads Priyadarsini et al. (2025); `ibm/materials.selfies-ted`.

- `SELFormer`: a RoBERTa-style encoder-only model for SELFIES, with 86M parameters, 12 layers, and 4 attention heads Yüksel et al. (2023). `HUBioDataLab/SELFormer`

- `MLT-BERT`: a BERT-style transformer model for sequence modeling, with 16M parameters, 8 layers, and 8 attention heads Zhang et al. (2022). `jonghyunlee/ChemBERT_ChEMBL_pretrained`

- `ChemBERTa`: a RoBERTa-style transformer encoder pretrained on 100,000 SMILES strings from the ZINC database using masked language modeling. The model consists of 6 transformer layers, 12 attention heads per layer (72 attention mechanisms in total), and a hidden dimension of 768 Chithrananda et al. (2020). `seyonec/ChemBERTa-zinc-base-v1` checkpoint.

- `GROVER`: a hybrid transformer-GNN model, with 48M ($\text{GROVER}_{\text{base}}$) and 100M ($\text{GROVER}_{\text{large}}$) parameters, both using 85 functional groups as the motifs of molecules. The graph-level embedding is obtained by concatenating a [CLS] token with pooled atomic embeddings Rong et al. (2020). `gmum/huggingmolecules`.

For these global text and graph embedding models, denoted $\phi_{\text{LLM}}$ or $\phi_{\text{GNN}}$, we evaluated three kernel functions: linear, isotropic Gaussian, and isotropic Laplacian.

### B.1.3 Global Cartesian Coordinates Molecular Representations

**Coulomb Matrix (CM) Rupp et al. (2012)**: CM is a global descriptor that encodes pairwise electrostatic interactions between atoms:

$$C_{ij} = \begin{cases} 0.5 Z_i^{2.4} & \text{if } i = j \\ \frac{Z_i Z_j}{R_{ij}} & \text{if } i \neq j, \end{cases} \tag{12}$$

where $Z_i$ is the atomic number of atom $i$ and $R_{ij}$ is the interatomic distance; $R_{ij} = \|\mathbf{R}_i - \mathbf{R}_j\|$. Despite its simplicity, CM is not invariant to atom indexing, which limits its generalization. To ensure invariance to atom indexing, each molecule is represented via the eigenvalue spectrum of its CM, sorted by descending absolute value. This diagonalized form is invariant to permutations, translations, and rotations, and yields a continuous molecular distance metric even for molecules with different numbers of atoms (using zero-padding).

**Bag of Bonds (BoB) Hansen et al. (2015)**: The BoB descriptor is inspired by the bag-of-words model from natural language processing, yielding rotational, translational, and permutation-invariant molecular representations. BoB extends the Coulomb Matrix by grouping pairwise atomic interactions into "bags" based on bond types, with each entry computed as $Z_i Z_j / |R_i - R_j|$. The entries in each bag are sorted by magnitude and zero-padded for consistent vector length. While effective for machine learning tasks, BoB cannot distinguish between homometric molecules.

**Spectrum of London and Axilrod–Teller–Muto (SLATM) Huang & von Lilienfeld (2020)**: SLATM builds on many-body expansions to describe molecular structures. It models atomic environments by approximating charge densities with Gaussian functions scaled by interatomic potentials. The representation captures one-body (atomic type), two-body (pairwise distances via a London-like potential), and three-body (angles via the Axilrod–Teller–Muto potential) interactions. Each term is binned into histograms to produce fixed-length atomic vectors, ensuring invariance to translation, rotation, and permutation. SLATM supports both local (atomic-level) representations and global ones formed by summing over atomic vectors, making it effective for a wide range of molecular machine learning tasks.

### B.2 Local Molecular Kernels

We considered three widely used local molecular kernels:

- **Smooth Overlap of Atomic Positions (SOAP)** Bartók et al. (2013).

- **Faber–Christensen–Huang–Lilienfeld 2019 (FCHL19)** Christensen et al. (2020).

- **Atom-Centered Symmetry Functions (ACSF)** Behler (2011).

As in prior works Faber et al. (2018); Christensen et al. (2020); Khan et al. (2023); Khan & von Lilienfeld (2024), local kernels are constructed using an element-matching function,

$$g(Z_i, Z_j) = \delta(Z_i = Z_j),$$

so that, in **Definition** 4, only atoms of the same chemical species in molecules $\mathcal{M}_i$ and $\mathcal{M}_j$ contribute to the kernel evaluation. In our experiments, all Gaussian and Laplacian kernels built on local representations follow this convention. The resulting local kernel takes the form

$$k_{\text{local}}(\mathcal{M}_i, \mathcal{M}_j) = \sum_{\ell_i=1}^{\text{Na}_i} \sum_{\ell_j=1}^{\text{Na}_j} \delta(Z_{\ell_i} = Z_{\ell_j}) \, \kappa\big(\phi_\ell(\boldsymbol{z}_{\ell_i}), \phi_\ell(\boldsymbol{z}_{\ell_j})\big), \tag{13}$$

where $\phi_\ell(\boldsymbol{z}_\ell)$ denotes the local atomic descriptor (e.g., SOAP, FCHL19, or ACSF), and $\kappa$ is either an isotropic Gaussian or Laplacian kernel.

### B.3 Implementation

In terms of implementation, ECFP representations were generated using `RDKit`, specifically ECFP4 (radius = 2, dimension = 1,024) and ECFP6 (radius = 3, dimension = 2,048). FCHL19, SLATM, CM, and BOB representations were computed via the `qml2` library. The ACSF and SOAP descriptors were generated using the `DScribe` package Himanen et al. (2020); Laakso et al. (2023). For SOAP, we utilized Gaussian-type radial basis functions with $r_{\text{cut}} = 6.0$ Å, $n_{\text{max}} = 3$, $l_{\text{max}} = 3$, and $\sigma = 0.1$. For ACSF, we employed a 6.0 Å cutoff with three $G^2$ radial symmetry functions and four $G^4$ angular terms.

## C Additional Results

**Datasets** In this paper, we consider the molecular property prediction as regression task on two types of molecular datasets:

1. **QM9** dataset Ramakrishnan et al. (2014), a benchmark of ∼134,000 small organic molecules containing up to nine heavy atoms (C, O, N, F). The molecular properties in QM9 were computed using density functional theory at the B3LYP/6-31G(2df,p) level. Our experiments focus on predicting seven core properties: the HOMO–LUMO gap (Gap), internal energy at 0 K ($U_0$) and 298.15 K ($U_{298}$), heat capacity ($C_V$), enthalpy ($\Delta H$), Gibbs free energy ($G$) at 298.15 K, and zero-point vibrational energy (ZPVE).

2. **MoleculeNet benchmark** Wu et al. (2018) consists of three regression tasks: ESOL, FreeSolv, and Lipophilicity. MoleculeNet aggregates multiple public datasets to provide standardized ML benchmarks for molecular properties. ESOL contains experimental aqueous solubility values, while FreeSolv provides hydration free energies derived from alchemical calculations and experiments. The Lipophilicity dataset, curated from the ChEMBL database, contains experimental octanol/water distribution coefficients (logP at pH 7.4) for approximately 4,200 molecules. Notably, molecules in these three datasets are provided solely as SMILES strings, enabling us to assess kernel performance in a regime that relies exclusively on 2D structural representations.

**Molecular Features as Models** Our experiment involves kernel features implicitly defined by 13 ECFP kernels, 3 3D-global kernels and 3 3D-local kernels, and pre-trained features explicitly extracted by 4 transformer-based encoders and 1 GNN-encoder:

1. **ECFP kernels** We utilize the ECFP global representation to construct 11 domain-specific molecular kernels (Tanimoto, Dice, Otsuka, Sogenfrei, Braun-Blanquet, Faith, Forbes, Inner-Product, Intersection, Min-Max, and Rand; see Section B.1.1 for mathematical details). Additionally, we apply standard Gaussian and Laplacian kernels directly to these ECFP representations as one-hot vectors.

2. **Pre-trained features:** We extract pre-trained features from state-of-the-art molecular transformers (SELFIESTED, SELFormer, ChemBERTa, and MLT-BERT) using string-based inputs (SMILES/SELFIES), as well as graph-level embeddings using a GNN (GROVER). We then construct Gaussian, Laplacian, and linear kernels on top of these extracted pre-trained features (see Section B.1.2).

3. **Global 3D representations:** We employ representations that capture the complete molecular geometry—namely the Coulomb matrix (CM), bag of bonds (BOB), and SLATM. Isotropic Gaussian, Laplacian, and linear kernels are built on top of these representations (see Section B.1.3).

4. **Local 3D representations:** We consider local structural descriptors that encode pairwise atomic environments, including local SOAP Bartók et al. (2013), FCHL19 Christensen et al. (2020), and ACSF Behler (2011). In practice, linear kernels are rarely used for local 3D representations; therefore, we pair them with nonlinear similarity measures (e.g., Gaussian or Laplacian kernels) to better capture the smooth geometric variations in atomic environments (see Section B.2).

**Hyperparameter Choice for QM9 dataset** The hyperparameters associated with the molecular representations were kept fixed to ensure consistency in the representations. For Gaussian and Laplacian kernels, the

length scale parameter $\ell$ was optimized via grid search over the set $\{0.1 \cdot 2^i \mid i = 0, \ldots, 14\}$, while for global kernels it was $\{10^n \mid n = 2, \ldots, 8\}$.. The regularization hyperparameter $\lambda$ was tuned separately for each representation through grid search combined with 4-fold cross-validation. Specifically, for fingerprint-based kernels, $\lambda$ was selected from the range $\{10^i \mid i = -10, \ldots, 2\}$, while for all other kernels, it was selected from $\{10^{-3i} \mid i = 1, \ldots, 9\}$. The best configuration was then selected based on the lowest mean absolute error (MAE) on the validation set.

### C.1 Kernel eigenvalue spectra

Figs. 8–11 are the eigenvalue spectra of various global and local kernels.

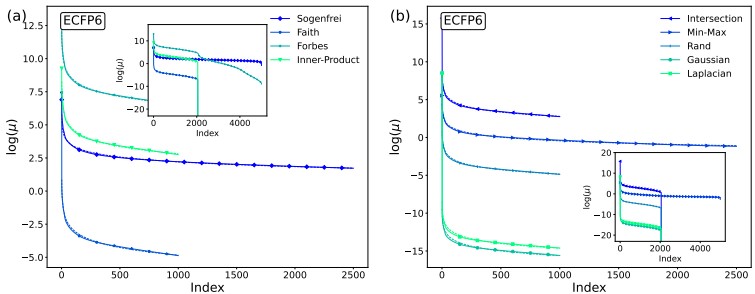

Figure 8: Kernel eigenvalue spectra with insets highlighting that nearly half of the eigenvalues are close to zero (main plots) for different ECFP-based kernels.

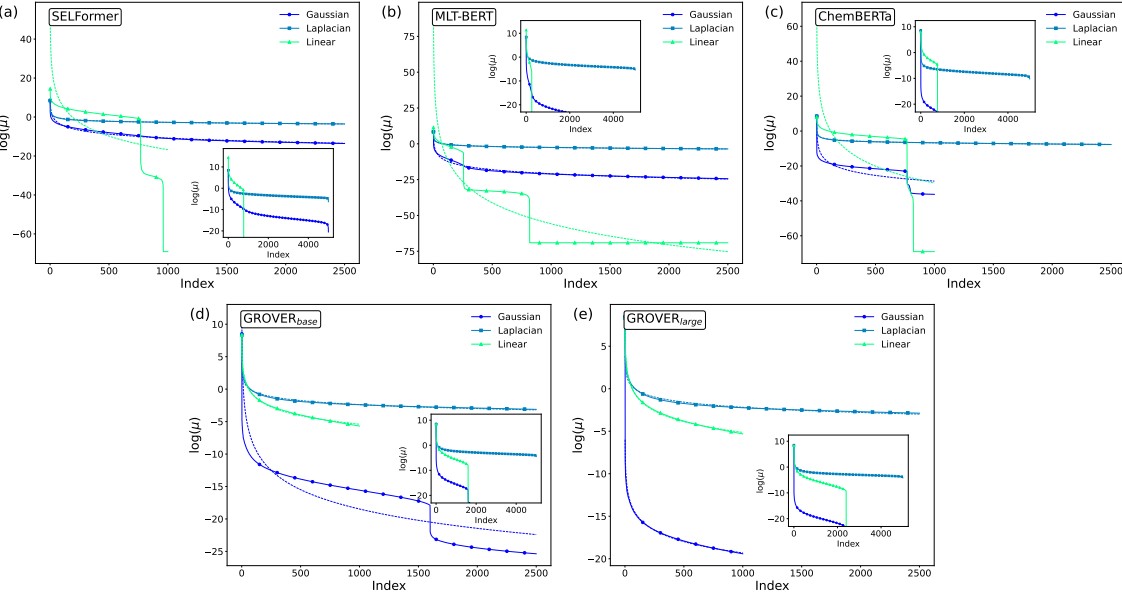

Figure 9: Kernel eigenvalue spectra with insets highlighting that nearly half of the eigenvalues are close to zero (main plots). For all we considered the Gaussian, Laplacian, and linear kernels.

### C.2 Truncation versus no Truncation

Figs. 12–14 represent the MAE, for a test set of $10,000$ molecules, for various properties when different truncation levels are considered. At each truncation level, all hyperparameters were optimized. Fig. 12 presents the results for four ECFP-based kernels , Fig. 13 for four global representations (CM, BOB, SELFIESTED, and MLT-BERT), all using the Gaussian kernel, and Fig. 14 for three local representations (SOAP, FCHL19, ACSF), all using the Gaussian kernel.

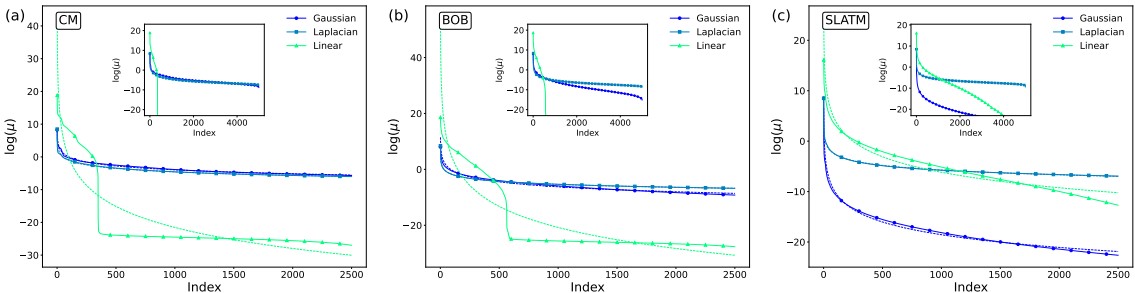

Figure 10: Kernel eigenvalue spectra with insets highlighting that nearly half of the eigenvalues are close to zero (main plots). For all, we considered the Gaussian, Laplacian, and linear kernels.

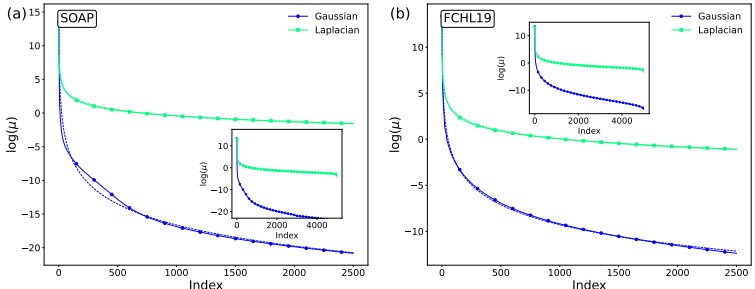

Figure 11: Kernel eigenvalue spectra with insets highlighting that nearly half of the eigenvalues are close to zero (main plots). For both, we considered the Gaussian and Laplacian kernels.

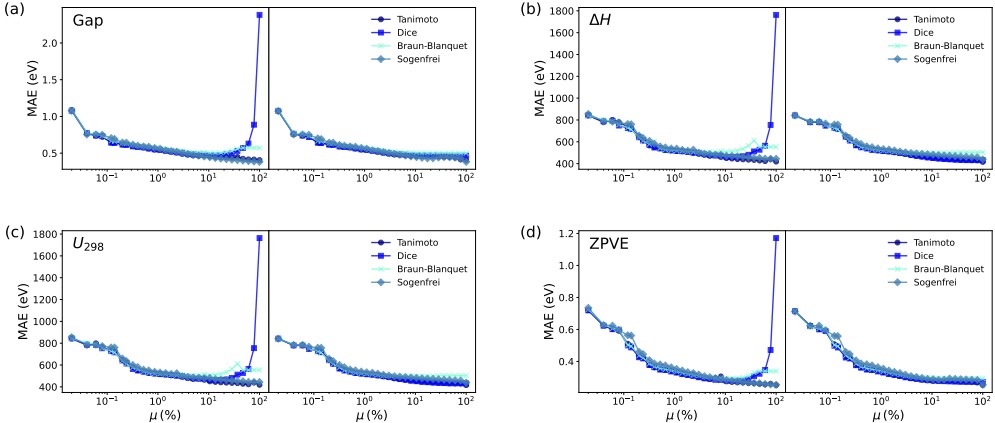

Figure 12: MAE for various properties as a function of truncation level for selected ECFP-based kernels. Left and right subpanels only consider results without and with regularization, respectively.

### C.3 Results for ECFP4 Fingerprint on QM9 dataset

Table 4 reports the spectral kernel metrics ($\alpha$, SSE, ID, SR) and MAEs across seven QM9 molecular properties for ECFP4 representation with all fingerprint kernels considered in this study.

### C.4 MoleculeNet Benchmark Datasets

For the ESOL, FreeSolv, and Liphophilicity datasets from the MoleculeNet benchmark Wu et al. (2018), data were randomly split into 80% training and 20% test sets, and this procedure was repeated four times with different random seeds. The regularization hyperparameter $\lambda$ and the length-scale parameter $\ell$ were

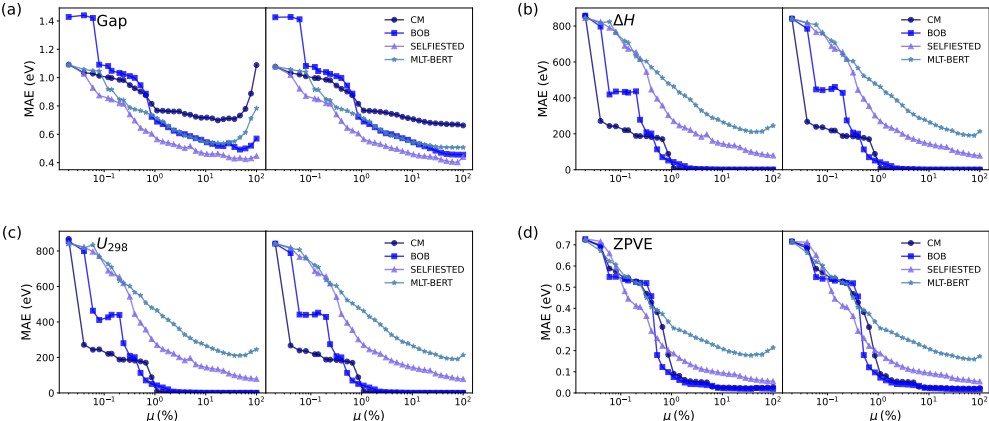

Figure 13: MAE for various properties as a function of truncation level for four global representations using Gaussian kernel. Left and right subpanels only consider results without and with regularization, respectively.

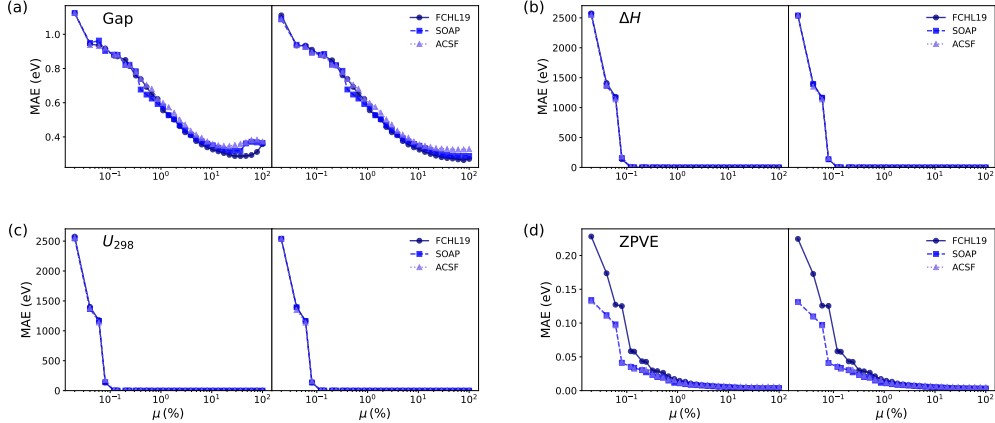

Figure 14: MAE for various properties as a function of truncation level for the three local representations using local Gaussian kernel. Left and right subpanels only consider results without and with regularization, respectively.

Table 4: Comparison of spectral metrics and MAE obtained from KRR using ECFP4 representation. Spectral metrics are reported for kernels with hyperparameters tuned on the $C_V$ property. The best and second-best MAE values for each property are highlighted in blue and red, respectively. The four spectral metrics quantify the richness of the kernel spectrum (direction indicated by arrows).

| Mol. Rep. | Kernel | $\alpha \downarrow$ | SSE $\uparrow$ | ID $\uparrow$ | SR $\uparrow$ | MAE $\downarrow$ | | | | | | |
|---|---|---|---|---|---|---|---|---|---|---|---|---|
| | | | | | | Gap (eV) | $C_V$ (cal/molK) | $\Delta H$ (eV) | $U_0$ (eV) | $U_{298}$ (eV) | $G$ (eV) | ZPVE (eV) |
| ECFP4 | Tanimoto | 0.85 | 1271.10 | 11.74 | 1.30 | 0.394 | 1.557 | 437.174 | 437.180 | 437.174 | 437.191 | 0.259 |
| | Dice | 0.93 | 248.39 | 6.57 | 1.24 | 0.482 | 1.645 | 455.062 | 455.068 | 455.062 | 455.077 | 0.287 |
| | Otsuka | 0.93 | 244.77 | 6.52 | 1.23 | 0.493 | 1.691 | 471.640 | 471.647 | 471.640 | 471.657 | 0.294 |
| | Sogenfrei | **0.64** | **2467.07** | **30.62** | **1.90** | 0.378 | 1.601 | 458.245 | 458.252 | 458.245 | 458.264 | 0.259 |
| | Braun-Blanquet | 0.94 | 241.10 | 6.47 | 1.23 | 0.516 | 1.745 | 511.511 | 511.517 | 511.511 | 511.528 | 0.303 |
| | Faith | 0.97 | 1.41 | 1.03 | 1.00 | 0.503 | 1.692 | 472.206 | 472.212 | 472.206 | 472.222 | 0.299 |
| | Forbes | 0.93 | 248.39 | 6.57 | 1.24 | 0.492 | 1.686 | 468.320 | 468.327 | 468.320 | 468.337 | 0.292 |
| | Inner-Product | 0.94 | 241.10 | 6.47 | 1.23 | 0.516 | 1.755 | 503.874 | 503.880 | 503.874 | 503.889 | 0.302 |
| | Intersection | 0.97 | 1.21 | 1.02 | 1.00 | 0.503 | 1.693 | 472.456 | 472.462 | 472.456 | 472.471 | 0.299 |
| | Min-Max | 0.85 | 1271.10 | 11.74 | 1.30 | 0.394 | 1.557 | 437.174 | 437.180 | 437.174 | 437.191 | 0.259 |
| | Rand | 0.97 | 1.21 | 1.02 | 1.00 | 0.503 | 1.692 | 472.206 | 472.212 | 472.206 | 472.221 | 0.299 |
| | Gaussian | 1.07 | 1.00 | 1.00 | 1.00 | 0.428 | 1.692 | 468.897 | 468.912 | 468.915 | 468.955 | 0.302 |
| | Laplacian | 1.04 | 1.00 | 1.00 | 1.00 | 0.413 | 1.675 | 467.336 | 467.342 | 467.336 | 467.352 | 0.283 |

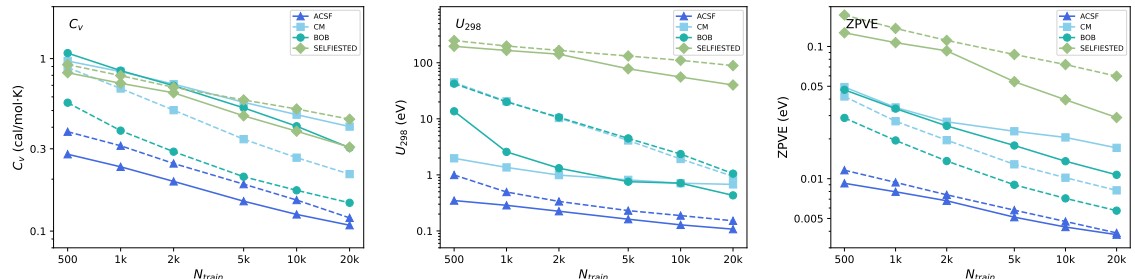

Figure 15: Test MAE, computed on $10,000$ molecules, as a function of training set size for three properties: (a) $C_V$, (b) $U_{298}$, and (c) ZPVE. Solid: Gaussian, and Dashed: Laplacian kernel.

optimized by grid search. For Gaussian and Laplacian kernels, $\ell$ was selected from $\{10^n \mid n = -2, \ldots, 7\}$. The regularization parameter was selected from $\{10^{-3n} \mid n = 1, \ldots, 9\}$ for continuous kernels and from $\{10^i \mid i = -10, \ldots, 2\}$ for fingerprint-based kernels. The best combination of hyperparameters was selected based on the model achieving the lowest MAE on the test set. In addition to the ECFP fingerprints, we evaluated transformer-based representations, including GROVER, SELFIESTED, SELFormer, ChemBERTa and MLT-BERT, and the results are presented in Table 5.

## C.5 Ablation Study

To investigate how different molecular representations contribute to the structure and stability of global kernels, we performed a comprehensive feature ablation study on the QM9 dataset. Using training and test sets of 5,000 and 10,000 molecules, respectively. Tables 6 and 7 report the test MAE for four properties ($C_V$, HOMO–LUMO gap, $U_0$, and ZPVE), together with the spectral metrics $\alpha$, SSE, SR, and ID, across different ablation levels. We consider three molecular representations: ECFP6, BOB, and SELFIES-TED. For ECFP6, results are reported for the Tanimoto, Dice, Gaussian, and Laplacian kernels, with $\ell \in \{10^n \mid n \in \{2, 4\}\}$.

**Ablation on Training Size** While the results in Table 2 are reported with a fixed training size of $N_{\text{train}} = 5,000$, we also conducted an ablation study varying $N_{\text{train}}$. Fig. 15 plots the mean absolute error across different kernels and molecular properties. The results show a steady improvement in test performance as $N_{\text{train}}$ increases to 10,000 and 20,000.

To examine whether spectral characterization changes with dataset size, Fig. 16 reports kernel metrics as a function of $N_{\text{train}}$. In contrast to the MAE, the kernel metrics show only modest variation across training sizes, indicating that the spectral structure of the kernels is an intrinsic property of the molecular representation, rather than by the specific training size used in the analysis. In particular, metrics such as SR remain

Table 5: Kernel metrics and MAE on three molecular property benchmarks. Bold: best spectral metric within each representation category. Blue / Red: best / second-best MAE within each category. Underlined digits reflect uncertainty at the std scale.

| Mol. Rep. | Kernel | $\alpha \downarrow$ | SSE $\uparrow$ | ID $\uparrow$ | SR $\uparrow$ | MAE $\downarrow$ |
|---|---|---|---|---|---|---|
| **ESOL** | | | | | | |
| ECFP4 | Tanimoto | 0.76 | 327.31 | 10.90 | 1.62 | 0.764 |
| | Dice | 1.06 | 138.90 | 6.33 | 1.41 | 0.840 |
| | Gaussian | 1.26 | 1.02 | 1.00 | 1.00 | 0.959 |
| | Laplacian | 1.25 | 1.04 | 1.00 | 1.00 | 0.966 |
| ECFP6 | Tanimoto | **0.66** | **433.98** | **14.29** | **1.78** | 0.780 |
| | Dice | 0.88 | 216.92 | 8.07 | 1.50 | 0.805 |
| | Gaussian | 1.06 | 1.03 | 1.00 | 1.00 | 0.912 |
| | Laplacian | 0.83 | 16.37 | 1.57 | 1.00 | 0.890 |
| SELFIES-TED | Gaussian | 1.43 | **8.10** | **1.58** | 1.03 | 0.527 |
| | Laplacian | **0.95** | 4.13 | 1.20 | 1.00 | 0.520 |
| SELFormer | Gaussian | 2.12 | 2.57 | 1.14 | 1.01 | 1.066 |
| | Laplacian | 0.97 | 4.27 | 1.24 | 1.00 | 0.935 |
| MLT-BERT | Gaussian | 2.88 | 1.43 | 1.06 | 1.00 | 0.913 |
| | Laplacian | 1.14 | 6.87 | 1.49 | **1.03** | 0.868 |
| ChemBERTa | Gaussian | 1.85 | 5.96 | 1.38 | 1.02 | 0.865 |
| | Laplacian | 1.01 | 4.91 | 1.26 | 1.00 | 0.816 |
| GROVER$_{base}$ | Gaussian | 1.94 | 1.85 | **1.11** | 1.00 | 0.554 |
| | Laplacian | 1.19 | 1.00 | 1.00 | 1.00 | 0.577 |
| GROVER$_{large}$ | Gaussian | 1.89 | **1.86** | 1.11 | 1.00 | 0.584 |
| | Laplacian | **1.18** | 1.00 | 1.00 | 1.00 | 0.572 |
| **FreeSolv** | | | | | | |
| ECFP4 | Tanimoto | 0.74 | 217.39 | 10.91 | 1.86 | 1.109 |
| | Dice | 1.01 | 103.85 | 6.51 | 1.59 | 1.230 |
| | Gaussian | 1.88 | 1.00 | 1.00 | 1.00 | 1.346 |
| | Laplacian | 1.21 | 1.96 | 1.07 | 1.00 | 1.315 |
| ECFP6 | Tanimoto | **0.67** | **266.31** | **13.05** | **1.95** | 1.133 |
| | Dice | 0.90 | 141.04 | 7.63 | 1.63 | 1.159 |
| | Gaussian | 1.97 | 1.00 | 1.00 | 1.00 | 1.219 |
| | Laplacian | 0.94 | 6.58 | 1.33 | 1.00 | 1.188 |
| SELFIES-TED | Gaussian | 1.59 | 4.53 | 1.34 | 1.01 | 0.807 |
| | Laplacian | 1.69 | 1.00 | 1.00 | 1.00 | 0.948 |
| SELFormer | Gaussian | 2.17 | 1.07 | 1.01 | 1.00 | 2.095 |
| | Laplacian | 1.11 | 2.89 | 1.15 | 1.00 | 1.961 |
| MLT-BERT | Gaussian | 1.16 | **60.79** | **4.29** | **1.31** | 1.431 |
| | Laplacian | 1.09 | 5.51 | 1.32 | 1.00 | 1.531 |
| ChemBERTa | Gaussian | 2.43 | 1.01 | 1.00 | 1.00 | 1.663 |
| | Laplacian | **1.08** | 4.36 | 1.26 | 1.00 | 1.546 |
| GROVER$_{base}$ | Gaussian | 1.82 | **1.95** | **1.13** | 1.00 | 0.928 |
| | Laplacian | 1.73 | 1.00 | 1.00 | 1.00 | 1.115 |
| GROVER$_{large}$ | Gaussian | 1.77 | 1.93 | 1.12 | 1.00 | 0.940 |
| | Laplacian | **1.71** | 1.00 | 1.00 | 1.00 | 1.085 |
| **Lipophilicity** | | | | | | |
| ECFP4 | Tanimoto | 0.92 | 621.88 | 7.55 | 1.12 | 0.570 |
| | Dice | 2.10 | 156.80 | 4.34 | 1.07 | 0.693 |
| | Gaussian | 1.31 | 12.64 | 1.44 | 1.00 | 0.615 |
| | Laplacian | 1.08 | 60.58 | 2.06 | 1.00 | 0.592 |
| ECFP6 | Tanimoto | **0.79** | **988.46** | **10.52** | **1.16** | 0.579 |
| | Dice | 1.20 | 327.31 | 5.85 | 1.09 | 0.652 |
| | Gaussian | 1.00 | 34.25 | 1.71 | 1.00 | 0.616 |
| | Laplacian | 0.84 | 208.42 | 2.88 | 1.01 | 0.592 |
| SELFIES-TED | Gaussian | 1.38 | 9.38 | 1.52 | 1.01 | 0.623 |
| | Laplacian | 0.76 | 312.47 | 3.23 | 1.04 | 0.639 |
| SELFormer | Gaussian | 2.48 | 2.53 | 1.15 | 1.00 | 0.811 |
| | Laplacian | 0.65 | 297.63 | 3.46 | 1.03 | 0.765 |
| MLT-BERT | Gaussian | 3.61 | 1.20 | 1.02 | 1.00 | 0.915 |
| | Laplacian | 0.98 | 607.97 | **15.14** | **1.86** | 0.870 |
| ChemBERTa | Gaussian | 1.52 | 6.30 | 1.39 | 1.01 | 0.665 |
| | Laplacian | **0.63** | **667.87** | 5.45 | 1.05 | 0.645 |
| GROVER$_{base}$ | Gaussian | 2.11 | 1.29 | 1.04 | 1.00 | 0.533 |
| | Laplacian | **0.86** | **31.58** | **1.61** | **1.01** | 0.569 |
| GROVER$_{large}$ | Gaussian | 2.01 | 1.31 | 1.04 | 1.00 | 0.526 |
| | Laplacian | 0.91 | 2.40 | 1.11 | 1.00 | 0.552 |

nearly constant, while ID and $\alpha$ show only moderate fluctuations. Consequently, the trends observed at $N_{\text{train}} = 5{,}000$ appear to remain qualitatively consistent at larger scales.

Table 6: Test MAE and kernel metrics for ECFP6 across kernels, length scale $\ell$, and ablation level $\tilde{N}$. Bold MAE: best $\tilde{N}$ per $(\ell, \text{property})$ group.

| Rep | Kernel | $\sigma_\ell$ | $\tilde{N}$ | $\alpha$ | SSE | ID | SR | Cv (cal/(mol·K)) | ZPVE (eV) | Gap (eV) | U0 (eV) |
|---|---|---|---|---|---|---|---|---|---|---|---|
| ECFP6 | Tanimoto | — | 0 | 0.71 | 1699.71 | 13.71 | 1.30 | **1.94** | **0.30** | **0.49** | **609.02** |
| | | | 16 | 0.70 | 1936.02 | 17.31 | 1.43 | 2.09 | 0.33 | 0.53 | 647.69 |
| | | | 64 | 0.68 | 2575.61 | 36.42 | 2.26 | 2.60 | 0.43 | 0.66 | 780.30 |
| | | | 512 | 0.71 | 3224.69 | 140.59 | 11.25 | 3.70 | 0.66 | 1.00 | 1103.68 |
| | Dice | — | 0 | 2.78 | 432.12 | 7.57 | 1.24 | **1.84** | **0.31** | **0.56** | **515.17** |
| | | | 16 | 2.70 | 520.98 | 9.44 | 1.35 | 1.92 | 0.33 | 0.57 | 532.60 |
| | | | 64 | 2.59 | 816.00 | 19.64 | 2.03 | 2.23 | 0.42 | 0.62 | 592.29 |
| | | | 512 | 2.87 | 1174.36 | 76.23 | 8.30 | 3.10 | 0.65 | 0.93 | 873.22 |
| | Gaussian | 100 | 0 | 2.99 | 1.03 | 1.00 | 1.00 | **1.63** | **0.28** | **0.48** | **456.46** |
| | | | 64 | 3.11 | 1.02 | 1.00 | 1.00 | 1.95 | 0.37 | 0.53 | 518.44 |
| | | | 256 | 3.31 | 1.02 | 1.00 | 1.00 | 2.23 | 0.47 | 0.69 | 645.57 |
| | | 1000 | 0 | 4.59 | 1.00 | 1.00 | 1.00 | **1.63** | **0.28** | **0.48** | **454.48** |
| | | | 64 | 4.73 | 1.00 | 1.00 | 1.00 | 1.99 | 0.37 | 0.54 | 521.90 |
| | | | 256 | 4.99 | 1.00 | 1.00 | 1.00 | 2.23 | 0.47 | 0.69 | 646.09 |
| | | 10000 | 0 | 4.82 | 1.00 | 1.00 | 1.00 | **1.63** | **0.28** | **0.48** | **454.28** |
| | | | 64 | 4.83 | 1.00 | 1.00 | 1.00 | 2.00 | 0.37 | 0.54 | 524.67 |
| | | | 256 | 4.88 | 1.00 | 1.00 | 1.00 | 2.24 | 0.48 | 0.69 | 647.46 |
| | Laplacian | 100 | 0 | 0.99 | 19.90 | 1.50 | 1.00 | **1.72** | **0.28** | **0.44** | **506.59** |
| | | | 64 | 1.04 | 13.66 | 1.38 | 1.00 | 2.04 | 0.37 | 0.51 | 562.81 |
| | | | 256 | 1.15 | 8.27 | 1.28 | 1.00 | 2.26 | 0.47 | 0.67 | 676.61 |
| | | 1000 | 0 | 1.45 | 1.54 | 1.04 | 1.00 | **1.63** | **0.28** | **0.48** | **460.37** |
| | | | 64 | 2.05 | 1.44 | 1.03 | 1.00 | 1.99 | 0.37 | 0.54 | 524.88 |
| | | | 256 | 2.22 | 1.33 | 1.02 | 1.00 | 2.23 | 0.47 | 0.68 | 647.66 |
| | | 10000 | 0 | 2.75 | 1.05 | 1.00 | 1.00 | **1.62** | **0.28** | **0.48** | **454.59** |
| | | | 64 | 2.87 | 1.05 | 1.00 | 1.00 | 1.99 | 0.37 | 0.54 | 522.07 |
| | | | 256 | 3.06 | 1.04 | 1.00 | 1.00 | 2.23 | 0.48 | 0.69 | 646.39 |

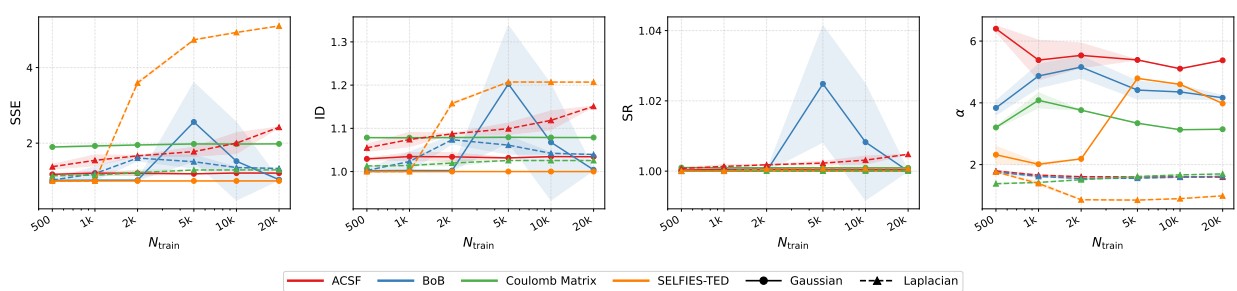

Figure 16: Kernel spectral metrics as a function of training size $N_{\text{train}}$ for different molecular representations and kernel types.

Fingerprint-based kernels show a more nuanced dependence on training size. Tanimoto kernel maintains stable spectral metrics and consistent MAE improvement with increasing $N_{\text{train}}$, whereas Braun-Blanquet and Dice show larger fluctuations in $\alpha$ and reduced stability at larger training sizes, suggesting a less robust spectral decay structure.

## D Spectral Metrics

**Definition 5** (Power Law Decay or polynomial decay rate). *Let* $\{\mu_1, \mu_2, \ldots, \mu_p\}$, $p \in \mathbb{N} \cup \{\infty\}$ *denote a non-increasing spectrum of positive values. We say the spectrum exhibits a* power law decay *if there exists an*

Table 7: Test MAE and kernel metrics for SELFIES-TED and BOB across kernels, length scale $\ell$, and ablation level $\tilde{N}$. Underlined digits mark the first uncertain decimal place. Bold MAE: best $\tilde{N}$ per $(\sigma_\ell, \text{property})$ group.

| Rep | Kernel | $\sigma_\ell$ | $\tilde{N}$ | $\alpha$ | SSE | ID | SR | Cv (cal/(mol·K)) | ZPVE (eV) | Gap (eV) | U0 (eV) |
|---|---|---|---|---|---|---|---|---|---|---|---|
| SELFIES-TED | Gaussian | 100 | 0 | 4.79 | 1.00 | 1.00 | 1.00 | **0.46** | **0.06** | 0.40 | **88.70** |
| | | | 64 | 4.85 | 1.00 | 1.00 | 1.00 | 0.46 | 0.06 | 0.40 | 90.68 |
| | | | 256 | 4.97 | 1.00 | 1.00 | 1.00 | 0.48 | 0.07 | 0.40 | 98.68 |
| | | | 512 | 4.89 | 1.00 | 1.00 | 1.00 | 0.54 | 0.08 | 0.41 | 119.76 |
| | | 1000 | 0 | 4.87 | 1.00 | 1.00 | 1.00 | **0.58** | **0.08** | 0.44 | **128.12** |
| | | | 64 | 4.86 | 1.00 | 1.00 | 1.00 | 0.59 | 0.09 | 0.44 | 131.78 |
| | | | 256 | 4.73 | 1.00 | 1.00 | 1.00 | 0.62 | 0.09 | 0.46 | 145.24 |
| | | | 512 | 4.21 | 1.00 | 1.00 | 1.00 | 0.69 | 0.11 | 0.48 | 170.78 |
| | | 10000 | 0 | 3.18 | 1.00 | 1.00 | 1.00 | **0.58** | **0.08** | 0.44 | **127.99** |
| | | | 64 | 3.16 | 1.00 | 1.00 | 1.00 | 0.59 | 0.09 | 0.44 | 131.55 |
| | | | 256 | 3.05 | 1.00 | 1.00 | 1.00 | 0.62 | 0.09 | 0.46 | 144.88 |
| | | | 512 | 2.71 | 1.00 | 1.00 | 1.00 | 0.69 | 0.11 | 0.48 | 170.71 |
| | Laplacian | 100 | 0 | 0.85 | 4.73 | 1.21 | 1.00 | **0.57** | **0.09** | 0.43 | **132.95** |
| | | | 64 | 0.86 | 4.37 | 1.19 | 1.00 | 0.58 | 0.09 | 0.44 | 135.57 |
| | | | 256 | 0.89 | 3.39 | 1.15 | 1.00 | 0.62 | 0.09 | 0.45 | 146.89 |
| | | | 512 | 0.95 | 2.35 | 1.10 | 1.00 | 0.67 | 0.11 | 0.46 | 169.20 |
| | | 1000 | 0 | 0.89 | 1.23 | 1.02 | 1.00 | **0.60** | **0.09** | **0.44** | **140.58** |
| | | | 64 | 0.90 | 1.22 | 1.02 | 1.00 | 0.60 | 0.09 | 0.45 | 143.71 |
| | | | 256 | 0.92 | 1.17 | 1.01 | 1.00 | 0.64 | 0.10 | 0.46 | 155.18 |
| | | | 512 | 0.98 | 1.12 | 1.01 | 1.00 | 0.70 | 0.11 | 0.47 | 177.69 |
| | | 10000 | 0 | 0.90 | 1.03 | 1.00 | 1.00 | **0.60** | **0.09** | **0.44** | **141.60** |
| | | | 64 | 0.91 | 1.02 | 1.00 | 1.00 | 0.61 | 0.09 | 0.45 | 147.07 |
| | | | 256 | 0.93 | 1.02 | 1.00 | 1.00 | 0.64 | 0.10 | 0.46 | 156.83 |
| | | | 512 | 0.99 | 1.01 | 1.00 | 1.00 | 0.70 | 0.12 | 0.47 | 178.55 |
| BOB | Gaussian | 100 | 0 | 2.56 | 9.23 | 2.07 | 1.13 | 0.49 | 0.02 | **0.45** | **11.50** |
| | | | 16 | 2.57 | 8.85 | 2.02 | 1.12 | **0.49** | **0.02** | 0.45 | 12.57 |
| | | | 64 | 2.63 | 8.18 | 1.97 | 1.12 | 0.50 | 0.02 | 0.46 | 11.67 |
| | | 1000 | 0 | 3.64 | 1.08 | 1.01 | 1.00 | **0.49** | **0.02** | 0.48 | **0.80** |
| | | | 16 | 3.65 | 1.07 | 1.01 | 1.00 | 0.49 | 0.02 | 0.48 | 5.34 |
| | | | 64 | 3.71 | 1.07 | 1.01 | 1.00 | 0.52 | 0.02 | 0.49 | 5.52 |
| | | 10000 | 0 | 4.82 | 1.00 | 1.00 | 1.00 | **0.58** | **0.02** | 0.51 | **0.77** |
| | | | 16 | 4.82 | 1.00 | 1.00 | 1.00 | 0.59 | 0.02 | 0.51 | 9.07 |
| | | | 64 | 4.85 | 1.00 | 1.00 | 1.00 | 0.62 | 0.02 | 0.52 | 9.12 |
| | Laplacian | 100 | 0 | 0.83 | 1154.36 | 28.43 | 4.43 | 1.24 | 0.12 | 0.50 | 415.45 |
| | | | 16 | 0.83 | 1144.26 | 27.98 | 4.34 | 1.23 | 0.12 | 0.49 | 408.66 |
| | | | 62 | 0.85 | 1039.90 | 25.85 | 4.19 | **1.16** | **0.11** | **0.49** | **377.70** |
| | | 1000 | 0 | 1.34 | 15.24 | 1.97 | 1.07 | **0.24** | **0.01** | 0.31 | 13.61 |
| | | | 16 | 1.34 | 14.92 | 1.95 | 1.07 | 0.24 | 0.01 | **0.31** | 13.71 |
| | | | 62 | 1.35 | 13.95 | 1.92 | 1.06 | 0.25 | 0.01 | 0.33 | **13.19** |
| | | 10000 | 0 | 1.54 | 1.65 | 1.08 | 1.00 | **0.21** | **0.01** | 0.33 | **4.52** |
| | | | 16 | 1.54 | 1.64 | 1.08 | 1.00 | 0.21 | 0.01 | 0.33 | 4.88 |
| | | | 62 | 1.55 | 1.62 | 1.08 | 1.00 | 0.22 | 0.01 | 0.34 | 5.43 |

exponent $\alpha > 0$ such that

$$\mu_j \propto j^{-\alpha}, \quad j = 1, 2, \dots, p. \tag{14}$$

The decay rate $\alpha$ can be estimated empirically by performing a linear regression on the log-log plot of the spectrum, i.e., $\log \mu_j \approx -\alpha \log j$ Agrawal et al. (2022); Mallinar et al. (2022).

**Definition 6** (p-Stable rank)**.** Suppose the integers $m \geq n$ and the matrix $\boldsymbol{A} \in \mathbb{R}^{m \times n}$ has singular values $s_1(\boldsymbol{A}) \geq s_2(\boldsymbol{A}) \geq \dots \geq s_n(\boldsymbol{A})$. For $1 \leq p \leq \infty$, the p-Schatten norm is defined to be

$$\|\boldsymbol{A}\|_p \overset{\text{def.}}{=} \sqrt[p]{s_1(\boldsymbol{A})^p + \dots + s_n(\boldsymbol{A})^p}. \tag{15}$$

*And the p-stable rank of the matrix $\boldsymbol{A}$ is defined to be*

$$r_p(\boldsymbol{A}) \stackrel{\text{def.}}{=} \frac{\|\boldsymbol{A}\|_p^p}{\|\boldsymbol{A}\|_{op}^p}. \tag{16}$$

**Definition 7** (Intrinsic dimension (ID) and stable rank (SR)). *Note that the notation of p-stable rank unifies the two metrics intrinsic dimension and stable rank, which are often used in ill-conditioned matrices. In particular, we have*

$$r_1(\boldsymbol{A}) = \frac{\|\boldsymbol{A}\|_1}{\|\boldsymbol{A}\|_{op}} = \frac{s_1(\boldsymbol{A}) + ... + s_n(\boldsymbol{A})}{s_1(\boldsymbol{A})} = \text{intrinsic dimension of } \boldsymbol{A};$$

$$r_2(\boldsymbol{A}) = \frac{\|\boldsymbol{A}\|_2^2}{\|\boldsymbol{A}\|_{op}^2} = \frac{\|\boldsymbol{A}\|_F^2}{\|\boldsymbol{A}\|_{op}^2} = \text{stable rank of } \boldsymbol{A}.$$

In particular, the true rank of $\boldsymbol{A}$ is always an upper bound of $r_p(\boldsymbol{A})$ for any $p$. In particular,

**Proposition 8** (Remark 5.4 in Ipsen & Saibaba (2024)). *Suppose the integers $m \geq n$ and $p \geq q$. Then for any matrix $\boldsymbol{A} \in \mathbb{R}^{m \times n}$, we have*

$$1 \leq r_p(\boldsymbol{A}) \leq r_q(\boldsymbol{A}) \leq \text{rank}(\boldsymbol{A}) \leq n. \tag{17}$$

We notice that there is another measure of rank used in ML literature:

**Definition 9** (Spectral Shannon Entropy (SSE), Definition 2.1 in Huh et al. (2023)). *Suppose the integers $m \geq n$ and the matrix $\boldsymbol{A} \in \mathbb{R}^{m \times n}$ has singular values $s_1(\boldsymbol{A}) \geq s_2(\boldsymbol{A}) \geq ... \geq s_n(\boldsymbol{A})$. Let $\bar{s}_i(\boldsymbol{A}) \stackrel{\text{def.}}{=} \frac{s_i(\boldsymbol{A})}{s_1(\boldsymbol{A}) + ... + s_n(\boldsymbol{A})}$ be the normalized singular values such that $\bar{s}_1(\boldsymbol{A}) + ... + \bar{s}_n(\boldsymbol{A}) = 1$. The spectral entropy, or the effective dimension, of $\boldsymbol{A}$ is defined to be:*

$$\rho(\boldsymbol{A}) \stackrel{\text{def.}}{=} \exp(-\sum_{i=1}^{n} \bar{s}_i(\boldsymbol{A}) \log(\bar{s}_i(\boldsymbol{A}))). \tag{18}$$

## E   Proof

In this section, we present the proof which are omitted in the main text.

KRR finds the optimal predictor $\hat{f}$ by minimizing the regularized empirical risk:

$$\hat{f}(\mathcal{M}) \stackrel{\text{def.}}{=} \min_{f \in \mathcal{H}} \sum_{i=1}^{N} (f(\mathcal{M}_i) - y_i)^2 + \lambda \|f\|_{\mathcal{H}}^2 = \sum_{i=1}^{N} \alpha_i k(\mathcal{M}_i, \mathcal{M}), \tag{19}$$

where $\alpha_i = [(\boldsymbol{K} + \lambda \boldsymbol{I})^{-1} \boldsymbol{y}]_i \in \mathbb{R}$, $\boldsymbol{K} \in \mathbb{R}^{N \times N}$ the kernel matrix, $\boldsymbol{y} \in \mathbb{R}^N$ is the target vector, and $\lambda \geq 0$ is the regularization constant.

**Theorem 10.** *With notation above, let $\hat{f}$ be the KRR predictor in Eq. (19) and $\hat{f}^{(r)}$ the TKRR predictor with truncation level $r$. Define*

$$\tilde{k}^{(r)}(\mathcal{M}_i, \mathcal{M}) = [\mathbf{U}_{\leq r} \mathbf{U}_{\leq r}^\top \mathbf{k}]_i \tag{20}$$

*where $\mathbf{U}_{\leq r} = (\mathbf{u}_k^\top)_{k=1}^r \in \mathbb{R}^{n \times r}$ is the sub-matrix of the orthonormal matrix $\mathbf{U} \in \mathbb{R}^{n \times n}$. Then we have*

1. *For any $r \leq n$ and $i, j$, $\tilde{k}^{(r)}(\mathcal{M}_i, \mathcal{M}_j) = K_{\mathcal{M}_i, \mathcal{M}_j}^{(r)}$ and hence $\tilde{f}^{(r)}(\mathcal{M}_i) = \hat{f}^{(r)}(\mathcal{M}_i)$ for all $i = 1, ..., n$.*

2. *For $r = n$ and any $i$ and any test point $\mathcal{M}$, $\tilde{k}^{(n)}(\mathcal{M}_i, \mathcal{M}) = k^{(n)}(\mathcal{M}_i, \mathcal{M})$ and hence $\tilde{f}^{(n)}(\mathcal{M}) = \hat{f}^{(n)}(\mathcal{M}) = \hat{f}(\mathcal{M})$.*

*Proof.* We use the standard notation in kernel theory: $\tilde{k}_{\boldsymbol{X},\boldsymbol{x}_i}^{(r)} = [\tilde{k}^{(r)}(\mathcal{M}_j, \mathcal{M}_i)]_{j=1}^n$, and analogously for $k_{\boldsymbol{X},\boldsymbol{x}_i}$. The first statement comes from:

$$\tilde{k}_{\boldsymbol{X},\boldsymbol{x}_i}^{(r)} = \mathbf{U}_{\leq r}\mathbf{U}_{\leq r}^\top K_{\boldsymbol{X},\boldsymbol{x}_i} = \mathbf{U}_{\leq r}\mathbf{U}_{\leq r}^\top \sum_{k=1}^n \mu_k u_{ik}\mathbf{u}_k = \sum_{k=1}^n \mu_k u_{ik}\mathbf{U}_{\leq r}\mathbf{U}_{\leq r}^\top\mathbf{u}_k = \sum_{k=1}^r \mu_k u_{ik}\mathbf{u}_k = k_{\boldsymbol{X},\boldsymbol{x}_i}^{(r)}.$$

The second statement comes from:

$$\tilde{k}_{\boldsymbol{X},\boldsymbol{x}}^{(n)} = \mathbf{U}\mathbf{U}^\top k_{\boldsymbol{X},\boldsymbol{x}} = K_{\boldsymbol{X},\boldsymbol{x}} = k_{\boldsymbol{X},\boldsymbol{x}}^{(n)}.$$

$\square$

In short, the above theorem establishes that for $r \leq n$ (1) our approximated TKRR predictor $\tilde{f}^{(r)}$ coincides with the TKRR predictor $\hat{f}^{(r)}$ on the training set, and (2) for $r = n$ it coincides with the original KRR predictor $\hat{f}$ on any new test points. As an independent contribution, this result extends the definition of TKRR beyond the original formulation in Amini et al. (2022), which may be of interest to kernel theorists.

## The Use of Large Language Models

In this work, we used large language models (LLMs) primarily as assistive tools to improve the clarity, grammar, and presentation of the manuscript. LLMs were employed to polish writing, rephrase sentences for readability, and ensure consistency in terminology. The use of LLMs did not influence the scientific content or conclusions of the paper; their role was limited to language refinement.

