# OpenReview forum: "Spectral Analysis of Molecular Features: When Richer Features Do Not Guarantee Better Generalization"
_TMLR — Under review for TMLR_

### Review · Reviewer_wT8c · 2026-07-12

**Summary Of Contributions:**

**Summary:**

This paper states that, based on a background that the spectral properties of features are widely used for property prediction, and their spectral behavior has not yet been explored. This work evaluates different backbones on QM9 and three molecule net benchmarks, giving a relatively comprehensive spectral analysis of the kernel and the previous embedding features. Beyond that, they have proposed Kernel Probing and Truncated Threshold of Features to improve the representations.

**Advantages:**

1. The motivation and original thinking of this paper is relatively good, which focuses on different features from the spectral side to consider and analyze the final contribution to the representations for a specific task.

2. This work also calculates the correlations between the final performance and the selected features, which is also meaningful for specific prediction situations and tasks. This direction, I think, is the right way to be optimal in real-world use cases and industrial scenarios.

**Disadvantages:**

1. More regression tasks should be chosen to prove the results. QM9 is a reasonable and standard benchmark to test; however, only three of the other benchmarks are not comprehensive in that case. I understand there aren't many regression tasks in the molecular net benchmark. That said, you should try to find several other regression benchmarks from TDC, for example, or try to use the classification task to validate your findings. By the way, I think in the representation tasks and property prediction scenarios, classification is also important and non-negligible, hence the analysis should also consider classification.

2. Many of the baselines are too old to compare. There are many new representation models for molecular embedding. The authors should consider using the newer models to validate their claims. Using only old models may not be suitable or may yield the same findings as the new models.

3. Most of the modern representation models are multi-task based, especially for the models that are self-supervised. Their motivation is to use a single supervised learning process, and then it can be adapted to downstream tasks effectively and efficiently. In this case, selecting more or fewer features is not that important, as they have their targets to handle many downstream tasks; there may not be optimal on a specific task, but their original goal is to be optimally average on a general-purpose embedding. If a model only focuses on a specific series of features, it may lose their generalization ability. This work should also analyze and discuss this issue. Should carefully reframe and distinguish their stands.

**Audience:**

Yes

**Audience Explanation:**

In my opinion, some people not want to make their prediction model specific on one task and make the prediction optimal and efficient enough, would be interested in this work. This work provides a method to select the most important spectral features to specific task, maybe inspiring to their work.

**Broader Impact Concerns:**

I didn't see any broader impact concerns.

**Claims And Evidence:**

No

**Claims Explanation:**

This work conducts enough experiments on several baselines based on their claim. It seems rigorous on the self.
However, the selection of baselines and datasets may not be comprehensive enough to support the core claim. The newer baseline should be tested instead of the older ones, which are more than 5 years old. Also, more benchmarks should be selected to test the results, especially in the classification scenarios.
Based on the reasons, I think the claims made by the submission are not supported by convincing evidence.

**Requested Changes:**

1. More molecular property prediction tasks. Only three additional MoleculeNet benchmarks are still not convicing enough. The authors should consider adding more regression datasets, for example, from TDC or other established benchmarks.

2. Classification tasks should also be included. In molecular representation learning and property prediction, classification is also important and non-negligible.

3. More recent molecular representation models should be added as baselines. The authors should verify whether the same findings still hold for modern representation models.

4. The proposed Kernel Probing and Truncated Threshold of Features should be evaluated on newer self-supervised and multi-task molecular models, rather than mainly on traditional or older backbones, too.

5. The authors should analyze the relationship between task-specific feature selection and general-purpose representation learning.

6. The authors should discuss whether selecting only a specific series of features may improve one downstream task but reduce the generalization ability on the others.

7. The paper should carefully distinguish task-specific optimization from general-purpose representation learning. The current claims should be reframed based on this.

---

### Review · Reviewer_W4eR · 2026-07-13

**Summary Of Contributions:**

This paper proposes a broad empirical spectral analysis of kernel ridge regression for molecular property prediction, spanning ECFP fingerprint kernels, pretrained transformer/GNN features, and global/local 3D descriptors, evaluated on QM9 and three MoleculeNet datasets. Using four spectral-richness metrics, the authors test the common SSL heuristic that "richer feature spectra yield better generalization." Their headline finding is that this heuristic does not hold universally in the molecular domain: correlations between spectral richness and performance depend strongly on the representation type. They additionally introduce "Kernel Probing" (KRR on frozen SSL features) and a truncated-KRR analysis quantifying how few eigenvalues are needed to recover most of the performance.

Strengths: extensive experiments are conducted. (20+ representation×kernel combinations, multiple datasets, four metrics); the framework unifying pretrained-feature and kernel-feature evaluation under one spectral axis is clean and potentially useful.
Weaknesses (detailed below): several claims are stated more strongly than the evidence supports, and two central results appear confounded by experimental-design choices (target preprocessing; correlating over a small number of kernels)

**Additional Comments:**

None.

**Audience:**

Yes

**Audience Explanation:**

A careful negative result challenging the "richer spectrum --> better generalization" heuristic, together with a broad spectral comparison of molecular kernels and pretrained features, is of clear interest to the kernel-methods, SSL-evaluation, and molecular-ML communities.

**Broader Impact Concerns:**

None.

**Claims And Evidence:**

Yes

**Claims Explanation:**

The qualitative direction of the main negative result is broadly consistent with the data, but several specific claims overreach, and two are confounded.

- Statistical power does not support the correlation claims. Each Pearson correlation in Table 3 is computed over only n = 6–13 points, where each point is one kernel (Global 3D: n=9, Local 3D: n=6). An n=6 Pearson is unstable, and most reported CIs include zero. Yet Table 1 and the text draw categorical conclusions such as "Consistently Positive/Negative." Calling Local 3D "consistently negative" while, in the same breath, conceding the wide CIs make it "statistically inconclusive" is internally contradictory.
- What is being tested is conceptually misaligned with the SSL heuristic. The SSL claim "richer features -> better generalization" concerns comparisons across different pretrained representations/models*. Here, within a representation category, the x-axis (spectral richness) is varied by applying different kernel functions to the same fixed features* (e.g., ECFP with Tanimoto/Dice...). This measures how the similarity function reshapes the spectrum, not whether a representation is richer — so the quantity actually varied does not match the heuristic the paper claims to test.
- The "few eigenvalues" result is likely confounded by the regression targets. In Table 2, the energy MAEs are 400 eV for ECFP thousands of times chemical accuracy (~0.043 eV)  and the U0/U298/ΔH/G columns are nearly identical (e.g., 415.975/415.982/415.975/415.992). This indicates the four "energy" targets are variants of a single total-energy target predicted *without atomization-energy baselining*. The most dramatic truncation result (Local 3D + energy targets recovering 95% with <2%, occasionally 0.02%, of eigenvalues) may therefore reflect that these targets are near-total-energy quantities dominated by molecular size — i.e., trivially low-dimensional — rather than any property of the representation.
- The Kernel Probing contribution is overstated. The paper frames KP as invented and as outperforming linear probing, but, as the authors themselves note, linear-kernel KP is linear probing; the reported gain is simply nonlinear (Gaussian/Laplacian) kernel regression outperforming linear regression on frozen features — an expected result, and kernel regression on frozen features is not itself new.
- A conceptual gap follows from the linear to nonlinear jump. The equivalence "l₂-regularized linear probing = linear-kernel KRR" holds only for the linear kernel. The paper immediately moves to nonlinear Gaussian/Laplacian kernels, whose KRR is no longer equivalent to SSL linear probing. This opens a gap between the stated motivation (testing the SSL heuristic) and what is actually measured (nonlinear-kernel spectra)

**Requested Changes:**

**[critical]** Scope the correlation claims to the statistics. Report n for each correlation, drop categorical language ("consistently") where CIs include zero, and explicitly acknowledge the low statistical power (n=6 for Local 3D). Conclusions should be stated as trends consistent-with, not established-by, the data.

**[critical]** Address the energy-target confound. Either predict atomization energies (standard baselining) or justify raw-energy targets, and disentangle the "few eigenvalues suffice" result from intrinsic target difficulty (e.g., report a size-only/linear baseline for the energy targets, or show the result on targets not dominated by molecular size). Also note that U0/U298/ΔH/G are near-duplicate targets and avoid effectively 4× weighting them in the averaged R².

**[critical]** Clarify what is being tested. Make explicit that spectral richness is varied via kernel choice on fixed features (not across representations), and reconcile this with the SSL heuristic being invoked. Address the linear vs. nonlinear-kernel gap in the linear-probing equivalence.

**[critical]** Reframe Kernel Probing honestly: nonlinear-kernel regression outperforming linear probing is expected, and kernelized regression on frozen features is not novel; soften the "invent"/contribution framing.

**[critical]** Clarify the meaning of "generalization" throughout the paper. As used here it refers to in-distribution, i.i.d. random-split test error, not the out-of-distribution / scaffold-split generalization that the molecular-ML community typically means. The current framing — e.g., "better generalization" and "label-limited scientific tasks" in the abstract and introduction — invites misreading; the intended notion should be stated explicitly and early.

**[strengthening]** Report at least scaffold-split results on the MoleculeNet datasets alongside the random-split results. Random splits are known to be optimistic on this benchmark (scaffold splits are the recommended evaluation), and whether the spectrum–performance relationship holds under distribution shift is more relevant to the molecular-ML audience — and is not addressed by the current i.i.d. setup.

**[strengthening]** Include at least one 3D message-passing GNN (e.g., SchNet or an equivariant GNN) as a frozen feature extractor in the spectral comparison. This is a standard and strong 3D baseline currently absent from the analysis and would substantially broaden the representation coverage.

**[strengthening]** Increase the number of points per correlation (e.g., more kernels/hyperparameter settings as samples) to make the correlation analysis more robust.

---

### Review · Reviewer_eyfP · 2026-07-16

**Summary Of Contributions:**

The paper presents an empirical spectral analysis of kernel ridge regression for molecular property prediction across ECFP fingerprints, pretrained molecular embeddings, and 3D molecular descriptors. It evaluates whether common spectral richness metrics, such as eigenspectrum decay, spectral entropy, intrinsic dimension, and stable rank, correlate with downstream performance on QM9 and MoleculeNet tasks. The main finding is that richer spectra do not consistently imply better generalization, with the relationship depending strongly on the representation and kernel family. The paper also studies truncated KRR, showing that for several molecular kernels, a small fraction of leading eigenvalues can recover most of the predictive performance.

The work is of some interest to the cheminformatics community, especially because it brings together molecular kernels, pretrained embeddings, and spectral diagnostics in a systematic comparison. However, several major technical issues weaken the conclusions, including possible test-set leakage in hyperparameter selection, statistically fragile correlation analyses, and unclear or possibly incorrect definitions of several fingerprint kernels.

**Audience:**

Yes

**Audience Explanation:**

The question of whether spectral properties of molecular kernels and pretrained molecular representations are informative for downstream generalization is likely to be of interest to some researchers working on molecular representation learning. In particular, the paper attempts to connect spectral diagnostics used in self-supervised learning with molecular property prediction, which is a potentially useful direction. However, in its current form, the practical value of the findings is strongly limited by technical issues in the experimental design and analysis, so I do not think the paper currently provides reliable or actionable conclusions.

**Broader Impact Concerns:**

No broader impact concerns identified.

**Claims And Evidence:**

No

**Claims Explanation:**

Several serious methodological issues undermine the paper’s conclusions.

- Appendix C.4 appears to select MoleculeNet hyperparameters using the lowest MAE on the test set, which is test leakage and invalidates the reported benchmark numbers in Table 5. A validation split or nested CV is required, with final test performance reported separately.
- Kernel hyperparameters confound the spectral metrics. Gaussian/Laplacian spectrum depends strongly on length scale, so the paper’s interpretation of “spectral richness” as a representation property is unsafe without fixed or standardized length-scale analysis.
- Using raw Gram/covariance spectra without centering and normalization means SSE/ID/SR can be dominated by trivial feature norm, molecular size, or kernel scale effects rather than meaningful diversity.
- Averaging R² across QM9 properties is questionable because many targets are highly related and overweights one type of property. This weakens claims about the quality of broad representation.
- The analysis lacks uncertainty quantification and statistical testing. Reported trends depend on train/test split, molecule subset, and hyperparameter selection. The paper should provide variability over repeated splits or bootstrap confidence intervals and test whether key differences are statistically meaningful.

Also, the manuscript overclaims novelty: the core contribution is mostly empirical and should be reframed as an application/extension of existing kernel probing and molecular GP/KRR ideas.

There are also notation issues, e.g., the denominator in Eq. (8) should be (\sqrt{|x_1||x_2|}) not (\sqrt{|x_1| + |x_2|}).

Because of these flaws, the claims are not supported by sufficiently convincing evidence.

**Requested Changes:**

Critical:
- Fix MoleculeNet hyperparameter selection to use a proper validation split or nested CV, then report final test performance separately.
- Re-analyze spectral metrics at fixed or standardized kernel length scales, and separate representation-induced spectrum effects from kernel tuning effects.
- Use centered and normalized Gram/covariance matrices or otherwise control for scale/offset effects in spectral diagnostics.
- Add uncertainty quantification and statistical testing for the main correlations, performance differences, truncation thresholds, and spectrum-performance trends.
- Reframe novelty claims to avoid overstating methodological contributions; clearly position the work relative to existing molecular GP/KRR and kernel probing literature.

Important:
- Clarify how targets were aggregated for QM9 and justify the choice of average R²; consider more balanced evaluation across diverse properties.
- Clarify kernel definitions and check whether “Count” in Table 1 is defined correctly.
- Add missing references to pretrained features in the main text.

Minor:
- Address writing issues such as awkward sentence starts and the “On the other hand” phrasing.
- Discuss why only ESOL, FreeSolv, and Lipophilicity were chosen from MoleculeNet, and whether other benchmarks were considered.
- Include tree-based baselines such as RF/GB since they are dominant in molecular property prediction in the introduction.